# MicroRNA-138-5p suppresses excitatory synaptic strength at the cerebellar input layer

Igor Delvendahl[1,2,3] (ID), Reetu Daswani[4,6], Jochen Winterer[4] (ID), Pierre-Luc Germain[4] (ID), Nora Maria Uhr[4], Gerhard Schratt[2,4] and Martin Müller[1,2,5] (ID)

[1] *Department of Molecular Life Sciences, University of Zurich (UZH), Zurich, Switzerland*
[2] *Neuroscience Center Zurich, Zurich, Switzerland*
[3] *Institute of Physiology, Faculty of Medicine, University of Freiburg, Freiburg, Germany*
[4] *Lab of Systems Neuroscience, Institute for Neuroscience, Department of Health Science and Technology, Swiss Federal Institute of Technology ETH, Zurich, Switzerland*
[5] *University Research Priority Program (URPP), Adaptive Brain Circuits in Development and Learning (AdaBD), University of Zurich, Zurich, Switzerland*
[6] *Present address: Sixfold Bioscience Ltd, Translation and Innovation Hub, London, UK*

Handling Editors: Katalin Toth & Samuel Young.

The peer review history is available in the Supporting Information section of this article (https://doi.org/10.1113/JP288019#support-information-section).

**Abstract figure legend** MicroRNAs influence neural function by regulating mRNA translation. We studied the role of microRNA 138 (miR-138-5p) in excitatory synaptic transmission at mossy fibre synapses of the mouse cerebellum. We downregulated endogenous miR-138-5p via a sponge construct and analysed synaptic function by patch-clamp electrophysiology. Our results show that downregulation of miR-138-5p increases excitatory postsynaptic currents (EPSCs) in cerebellar mossy fibre to granule cell synapses. The increase in synaptic strength is mediated by enhanced AMPA receptor function as well as increased neurotransmitter release probability (pr). miR-138-5p thus negatively regulates both presynaptic and postsynaptic function at the main excitatory input synapse to the cerebellar cortex.

**Igor Delvendahl** is a group leader at the Institute of Physiology, Albert-Ludwigs University of Freiburg, Germany. His research focuses on fundamental aspects of synaptic transmission and its impact on the function of neural circuits. Dr Delvendahl combines diverse methods including subcellular electrophysiology, two-photon imaging, machine learning and computational modelling to gain novel insights into synaptic physiology. With his scientific work, he aims to advance our understanding of the role of synapses for brain function and neural disorders.

**Abstract**  MicroRNAs are small, highly conserved non-coding RNAs that negatively regulate mRNA translation and stability. In the brain, miRNAs contribute to neuronal development, synaptogenesis, and synaptic plasticity. MicroRNA 138-5p (miR-138-5p) controls inhibitory synaptic transmission in the hippocampus and is highly expressed in cerebellar excitatory neurons. However, its specific role in cerebellar synaptic transmission remains unknown. Here, we investigated excitatory transmission in the cerebellum of mice expressing a sponge construct that sequesters endogenous miR-138-5p. Mossy fibre stimulation-evoked EPSCs in granule cells were ∼40% larger in miR-138-5p sponge mice compared to controls. Furthermore, we observed larger miniature EPSC amplitudes, suggesting an increased number of functional postsynaptic AMPA receptors. High-frequency train stimulation revealed enhanced short-term depression following miR-138-5p downregulation. Together with computational modelling, this suggests a negative regulation of presynaptic release probability. Overall, our results demonstrate that miR-138-5p suppresses synaptic strength through pre- and postsynaptic mechanisms, providing a potentially powerful mechanism for tuning excitatory synaptic input into the cerebellum.

(Received 31 October 2024; accepted after revision 14 April 2025; first published online 9 May 2025)

**Corresponding author** I. Delvendahl: Department of Molecular Life Sciences, University of Zurich (UZH), 8057 Zurich, Switzerland.    Email: igor.delvendahl@physiologie.uni-freiburg.de

## Key points

- MicroRNAs are powerful regulators of mRNA translation and control key cell biological processes including synaptic transmission, but their role in regulating synaptic function in the cerebellum has remained elusive.
- In this study, we investigated how microRNA-138-5p (miR-138-5p) modulates excitatory transmission at adult murine cerebellar mossy fibre to granule cell synapses.
- Downregulation of miR-138-5p enhances excitatory synaptic strength at the cerebellar input layer and increases short-term depression.
- miR-138-5p exerts its regulatory function through both pre- and postsynaptic mechanisms by negatively regulating release probability at mossy fibre boutons, as well as functional AMPA receptor numbers in granule cells.
- These findings provide insights into the role of miR-138-5p in the cerebellum and expand our understanding of microRNA-dependent control of excitatory synaptic transmission and short-term plasticity.

## Introduction

Synaptic transmission underlies a broad variety of brain functions ranging from sensory integration to learning and memory formation. The transmission efficacy of individual synapses is thus a key factor determining neural function. The efficacy or strength of synapses depends on their molecular composition and arrangement, as well as their history of activity, and is tightly controlled to enable proper functioning of the nervous system (Clayton et al., 2024; Grant, 2012; Zoghbi & Bear, 2012). Synapses can adjust their transmission efficacy not only to allow adaptive modifications (Bliss et al., 2013; Citri & Malenka, 2008), but also to counteract maladaptive deviations (Delvendahl & Müller, 2019). However, the mechanisms that allow such extensive and intricate control of synaptic strength are poorly understood.

The proteomic landscape of synapses is highly complex (Gonzalez-Lozano et al., 2020; Van Oostrum et al., 2023). Several key synaptic proteins have been attributed important functions for synaptic transmission or plasticity (Sheng & Kim, 2011; Südhof, 2012). Typically, the reduction or elimination of the expression of these proteins results in loss-of-function phenotypes, highlighting that many synaptic proteins facilitate synaptic transmission. Conversely, a subset of synaptic proteins acts to suppress synaptic strength by constraining neurotransmitter release and/or postsynaptic function, such as Tomosyn (McEwen et al., 2006), Mover (Körber et al., 2015) or SynGAP (Araki et al., 2015). Mechanisms that limit synaptic efficacy may be particularly important for maintaining robust information processing and preventing overexcitation

of neural networks (Eiro et al., 2023; Leite et al., 2005). However, our understanding of the negative regulation of synaptic strength remains limited.

MicroRNAs (miRNAs) are small non-coding RNAs that can bind to the $3'$ untranslated regions (UTRs) of multiple target mRNAs and suppress their translation (Filipowicz et al., 2008; Friedman et al., 2009). The biological functions of miRNAs are diverse. In the nervous system, miRNAs influence various processes ranging from neuronal development and synaptogenesis to memory formation and behaviour (Brennan & Henshall, 2020; Konopka et al., 2010; Kosik, 2006; Soutschek & Schratt, 2023). At the synaptic level, specific miRNAs can control the abundance of postsynaptic AMPA receptors (AMPARs) (Hanley, 2021) and affect synaptic plasticity (Schratt, 2009). Certain miRNAs are enriched in synapto-dendritic compartments (Sambandan et al., 2017; Schratt et al., 2006; Siegel et al., 2009), suggesting an important role in the local regulation of synaptic function. Among these, miR-138-5p regulates the morphology of dendritic spines and has a suppressive influence on synaptic function in primary rat hippocampal neurons (Siegel et al., 2009). On a behavioural level, loss of miR-138-5p impairs short-term memory in mice, which is most probably the result of an increase in inhibitory transmission in area CA1 of the hippocampus (Daswani et al., 2022). In inhibitory hippocampal interneurons, miR-138-5p may modulate synaptic transmission by inhibiting expression of the schizophrenia susceptibility gene *Erbb4* (Erb-B2 receptor tyrosine kinase 4) (Daswani et al., 2022). miR-138-5p is strongly expressed in the cerebellar cortex (He et al., 2012; Obernosterer et al., 2006). In this brain region, miRNAs have been shown to be important during development (Constantin, 2017), although their functions in the adult cerebellum remain largely unexplored.

The cerebellum is involved in a wide range of tasks, from motor control to cognitive functions (Carey, 2024; De Zeeuw et al., 2021; Schmahmann, 2004). Its computational capabilities depend on a highly conserved, canonical circuit architecture (Albus, 1971; Apps et al., 2018; Marr, 1969). Excitatory inputs to the cerebellar cortex are provided by climbing fibres (CFs) and mossy fibres (MFs). CFs directly target Purkinje cells (PCs), whereas MFs primarily synapse with cerebellar granule cells (GCs). GCs are highly numerous and provide an increase in dimensionality and sparsening of sensory information, which is essential for pattern separation (Cayco-Gajic & Silver, 2019) and supports cerebellum-dependent learning (Kita et al., 2021). Information transfer at the MF–GC synapse is thus highly relevant for cerebellar function. However, the mechanisms controlling synaptic strength at this key cerebellar synapse are not well understood and very little is known about the negative regulation of MF–GC transmission.

Based on its prominent cerebellar expression, we hypothesized that miR-138-5p suppresses excitatory synaptic strength in the cerebellum. We quantified synaptic transmission at cerebellar MF–GC synapses in acute mouse brain slices upon expression of a sponge construct that sequesters endogenous miR-138-5p. Our findings reveal that synaptic strength is increased in miR-138 sponge-expressing GCs, driven by enhanced postsynaptic AMPAR function and elevated presynaptic release probability. Thus, miR-138-5p suppresses synaptic efficacy at MF–GC synapses via both presynaptic and postsynaptic mechanisms and may comprise an important regulator of synaptic excitation at the cerebellar input layer.

## Methods

### Animals

Experiments were performed in C57BL/6NTac-Gt(ROSA)26So[tm2459(LacZ, antimir_138)Arte] ('miR-138 floxed') and 138-sponge[ub] ('mir-138 sponge') mice. Details of the generation of these animals have been reported previously (Daswani et al., 2022). Expression of the miR-138 sponge construct in the cerebellum was confirmed using lacZ staining. Animals were treated in accordance with national and institutional guidelines. All experiments were approved by the Cantonal Veterinary Office of Zurich (authorization no. ZH206/16 and ZH194/21). Mice were housed in groups of three to five per cage, with food and water available *ad libitum*. Recordings were performed in adult (aged 4–8 months) male mice using four miR-138 sponge animals and five littermate miR-138 floxed controls.

### Single-molecule fluorescence *in situ* hybridization (smFISH)

smFISH for miRNA detection on cerebellar slices was performed using the ViewRNA Tissue Assay Fluorescence Kit (Thermo Fisher Scientific, Waltham, MA, USA) with miRNA pretreatment. The protocol followed the manufacturer's instructions, with two modifications: an additional baking step of 13 min at 60°C, and a reduced protease treatment duration of 7 min. Next, 16-μm thick cerebellar slices were prepared using a cryostat (Leica, Wetzlar, Germany). Brain tissues were collected after perfusion of the animal with 4% paraformaldehyde, followed by a gradual increase in sucrose concentration to 30% over 3 days. The slices were stored in OCT at –80°C until sectioning. Images were acquired with a confocal laser scanning microscope (LSM 880; Zeiss, Oberkochen, Germany) at 10× magnification.

## miRNA target prediction and enrichment analysis

To predict miRNA targets, we took the longest annotated UTR of each protein-coding gene in the Ensembl GRCm38 release 99 annotation and scanned it for 7/8mer canonical sites using scanMiR (Soutschek et al., 2022). For enrichment analysis, we used the SynGO release 20231201 (Koopmans et al., 2019) and genes expressed at >2 log counts-per-million in granule cells as a background. We excluded genes without any SynGO annotation from the background. For each miRNA, we tested the 7/8mer predicted targets for over-representation (hypergeometric test) of all SynGO terms that had at least 10 annotated genes in the filtered universe. Reported are the top 10 most significant terms for miR-138-5p targets, all with false discovery rates below computing accuracy.

## Gene expression data

For expression of miR-138-5p across brain regions, we used data from the miRNATissueAtlas (Rishik et al., 2025). To analyse gene expression across brain regions, we used the single-nuclei RNA-sequencing (RNA-seq) data from Langlieb et al. (2023) (accession id: nemo:dat-aa0jwmj). Annotated metaclusters including the external cuneate nucleus, pontine grey, lateral reticular nucleus, vestibular nuclei and tegmental reticular nucleus were grouped to represent precerebellar nuclei that are major sources of mossy fibres in the cerebellum. For gene expression in neurons of the cerebellum, we used the single-nuclei RNA-seq data from Kozareva et al. (2021) (accession id: GSE165371), specifically using the P60 mice samples. In both cases, we produced pseudobulk samples from the quantification and cell annotations provided by the authors.

## LacZ staining

$\beta$-galactosidase activity was detected using the chromogenic substrate X-gal. Fresh frozen brain sections were fixed with formalin and after washing incubated in X-gal working solution at 37°C for 24 h. After incubation, slides were washed and mounted for imaging (https://ihcworld.com/2024/01/26/x-gal-staining-protocol-for-beta-galactosidase).

## Electrophysiology

Mice were killed by rapid decapitation according to national guidelines. The cerebellar vermis was removed quickly and mounted using superglue in a chamber filled with cooled extracellular solution. Parasagittal 300 μm thick slices were cut using a Leica VT1200S vibratome (Leica), transferred to an incubation chamber at 35°C for 30 min and then stored at room temperature until experiments. The extracellular solution (artificial cerebrospinal fluid, ACSF) for slice cutting and storage contained (in mM): 125 NaCl, 25 $NaHCO_3$, 20 D-glucose, 2.5 KCl, 2 $CaCl_2$, 1.25 $NaH_2PO_4$ and 1 $MgCl_2$, aerated with 95% $O_2$ and 5% $CO_2$.

Cerebellar slices were visualized using an upright microscope equipped with a 60×, 1 NA water immersion objective, infrared optics and differential interference contrast (Scientifica, Uckfield, UK). The recording chamber was continuously perfused with ACSF supplemented with 10 μM D-APV, 10 μM bicuculline and 1 μM strychnine to isolate AMPAR-mediated transmission. Voltage clamp recordings were performed using a HEKA EPC10 USB patch clamp amplifier (HEKA Elektronik GmbH, Lambrecht, Germany) and Patchmaster software (HEKA Elektronik GmbH; RRID:SCR_000034). Data were filtered at 10 kHz and digitized at 200 kHz; recordings of spontaneous postsynaptic currents were filtered at 2.7 kHz and digitized at 50 kHz. All experiments were performed at room temperature (22–25°C). Patch pipettes were pulled to open-tip resistances of 5–8 MΩ (when filled with intracellular solution) from borosilicate glass (Science Products, Hofheim am Taunus, Germany) using a DMZ puller (Zeitz Instruments, Munich, Germany). The intracellular solution contained (in mM): 150 K-D-gluconate, 10 NaCl, 10 Hepes, 3 MgATP, 0.3 NaGTP and 0.05 EGTA, pH adjusted to 7.3 using KOH. Voltages were corrected for a liquid junction potential of +13 mV.

Electrophysiological recordings were performed blinded to genotype. We recorded from visually identified GCs in lobules III–VI of the cerebellar vermis. Spontaneous miniature excitatory postsynaptic currents were recorded at a holding potential of –100 mV for 120–360 s. Extracellular MF stimulation was performed using bipolar square voltage pulses (duration, 150 μs) generated by an ISO-STIM 01B stimulus isolation unit (NPI, Tamm, Germany) and applied through an ACSF-filled pipette. The pipette was moved over the slice surface in the vicinity of the patched GC at the same time as applying voltage pulses until EPSCs could be evoked reliably. Care was taken to stimulate single mossy fibre inputs, as demonstrated by robust average EPSC amplitudes when increasing stimulation intensity (Silver et al., 1996). EPSC recordings were performed at a frequency of 0.1 Hz with stimulation intensity 1–2 V above the threshold, typically <20 V. For high-frequency train stimulation, we applied 20 pulses at 100 or 300 Hz, followed by six single pulses to monitor recovery from depression. Consecutive recovery pulses were spaced at inter-stimulus intervals of 25, 50, 100, 300, 1000 and 3000 ms. Train stimulations were repeated five times with 30 s interval.

Recordings from PCs were carried out in a similar fashion and using the same solutions. Recording pipettes

had a resistance of 2–5 MΩ, and the stimulation pipette was positioned in the molecular layer. Spontaneous EPSCs were recorded at a holding potential of –70 mV for 90–120 s in the presence of D-APV, bicuculline and strychnine. For action potential (AP)-evoked EPSCs, stimulation voltage was increased from 2–20 V in steps of 1 V, and 10 EPSCs were recorded at each step at –70 mV holding potential. The average EPSC amplitude for each step was plotted against the voltage and the slope determined by linear regression.

### Data analysis

Data analysis was performed blinded to genotype. Spontaneous mEPSCs were detected and analysed using *miniML* (O'Neill et al., 2025) with a model trained on cerebellar GC mEPSCs. For spontaneous EPSC detection in PCs, data were downsampled to 5 kHz before analysis. Evoked EPSCs were analysed using custom-written routines in IgorPro (WaveMetrics Inc., Portland, OR, USA; RRID:SCR_000325). EPSC amplitudes were measured as peak-to-baseline following 50 point binomial smoothing of the recordings. Baseline was averaged from a 2 ms window before stimulation, and peaks were calculated from a 100 μs window centred around the EPSC minimum. For train-stimulation EPSCs, a 400 μs baseline window after the stimulation artifact was used. EPSC charge was integrated over the entire duration of the EPSC, and decay times were extracted from biexponential fits to the decay of the averaged EPSC of each cell. AMPA receptor single-channel conductance was analysed from the variance of peak-scaled mEPSC decays (Traynelis et al., 1993) using recordings with >50 detected events as previously described (Delvendahl et al., 2019). For calculation of quantal content and RRP, mEPSC amplitudes were scaled according to the difference in holding voltage (–100 mV *vs.* –80 mV), assuming an AMPAR current reversal potential of 0 mV. Recovery from short-term depression was normalized to the first EPSC amplitude of the train stimulation. To quantify the time course of recovery, we fit averaged normalized EPSC amplitudes for each recording with a bi-exponential function. We only included synapses that reached at least 85% of the initial EPSC amplitude in the ∼5 s interval covered by the single recovery EPSC stimulations. Charge transfer during high-frequency train stimulation was calculated as cumulative area under each EPSC and its baseline current, which was measured immediately after the stimulation artifact. This analysis considers only the phasic EPSC component (Saviane & Silver, 2006). Traces shown in figures were filtered using a 15–25 samples Hann window for display purposes. Bars indicate mean values with error bars representing 95% confidence interval. Single data points represent individual cells, unless indicated.

### Computational modelling

We fit a Tsodyks–Markram model (Tsodyks & Markram, 1997; Tsodyks et al., 1998) to the MF–GC EPSC train data to extract the utilization of synaptic efficacy ($U_{SE}$). This model parameter represents the fraction of (presynaptic) resources used upon an action potential. The model parameters ($U_{SE}$, $A_{SE}$, $t_{facilitation}$ and $t_{recovery}$) were optimized for each recording using normalized EPSC amplitudes recorded at 100 and 300 Hz. We ran model optimization via an evolutionary algorithm (Blue Brain Python Optimisation Library) with offspring size = 500, $\eta$ = 20, mutation probability = 0.4 and cross-over probability = 0.7. Simulations were run for a maximum of 50 generations. Simulations were implemented in Python 3.9 (https://www.python.org/downloads/release/python-390/) using the *NEURON* (Hines, 2009) and *BluePyOpt* (Van Geit et al., 2016) libraries.

### Statistical analysis

We used the Python library *dabest* for statistical analyses (Ho et al., 2019). Effect sizes are reported as Cohen's $d$ with 95% confidence intervals, obtained by bootstrapping (5000 samples; the confidence interval is bias-corrected and accelerated) (Ho et al., 2019). Statistical comparisons were performed using permutation $t$ tests with 5000 reshuffles.

## Results

### miR-138-5p targets synaptic genes and is expressed in excitatory cerebellar and precerebellar neurons

To understand how miR-138-5p might regulate synaptic function in the cerebellum, we first examined the annotated functions of the strongest predicted miR-138-5p targets (i.e. 7/8mer) among genes expressed in cerebellar GCs using over-representation of SynGO terms (Koopmans et al., 2019). This analysis revealed enrichments for genes associated with both presynaptic or postsynaptic compartments (Fig. 1*A*). Given that synaptic genes tend to have a longer UTR, and as such are enriched for miRNA binding sites in general, we also established that these enrichments were much stronger for miR-138-5p targets than for most other miRNAs (Fig. 1*A*). This observation indicates that miR-138-5p may play an important role in regulating genes involved in pre- and postsynaptic function and suggests that miR-138-5p could modulate synaptic function in cerebellar GCs.

 miR-138-5p is a brain-enriched miRNA previously reported to be expressed in the hippocampus and the cerebellum based on *in situ* hybridization studies (Obernosterer et al., 2006). To corroborate these findings,

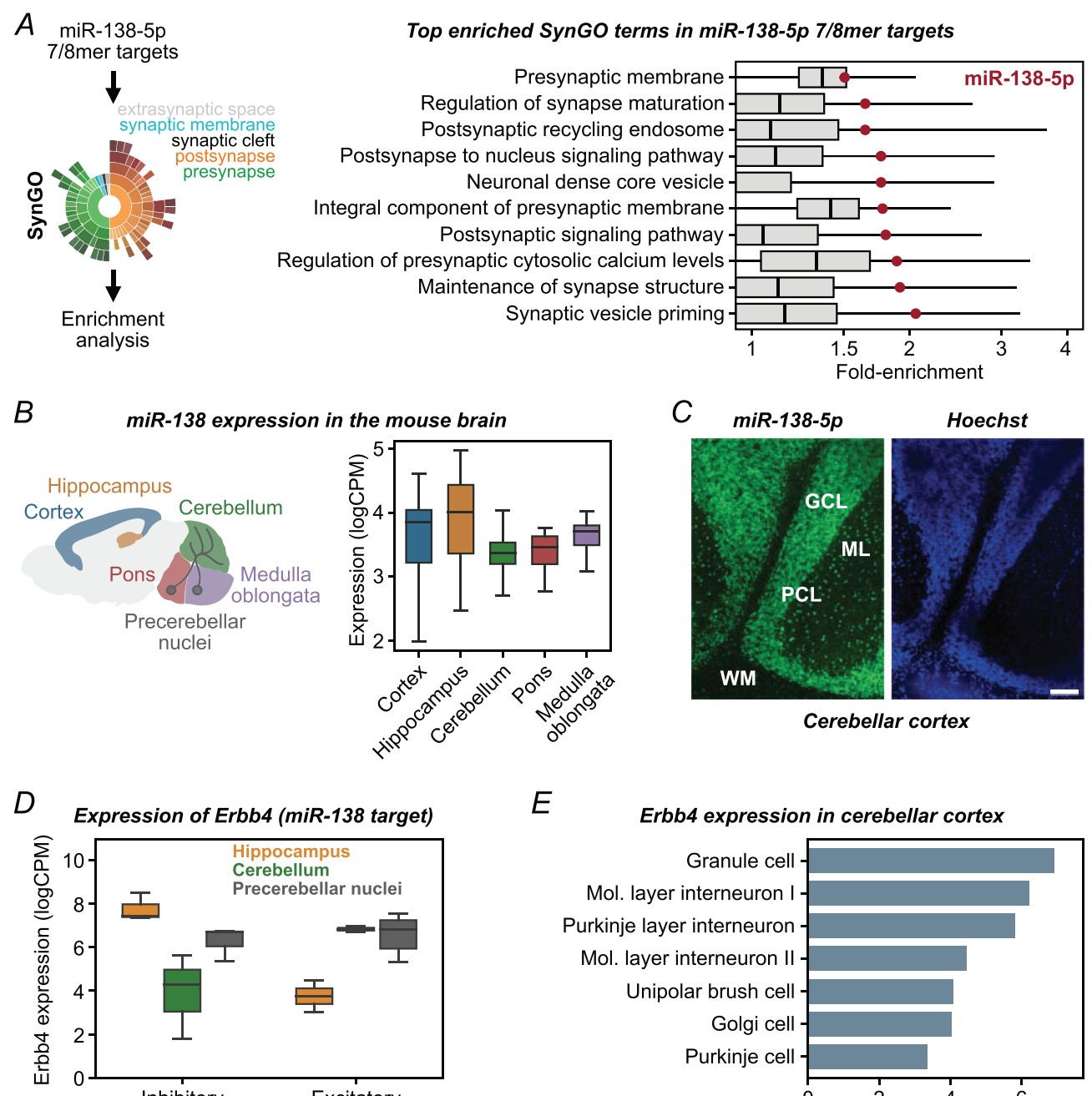

**Figure 1. miR-138-5p targets synaptic genes and is expressed in excitatory cerebellar and precerebellar neurons**

*A*, top 10 SynGO terms significantly enriched in the 7/8mer targets of miR-138-5p. The targets of many miRNAs are enriched for pre-/post-synaptic genes, whereas miR-138-5p (red dots) is consistently amongst those showing strongest enrichment. *B*, expression of miR-138-5p in different brain regions. miR-138-5p shows expression in the cerebellum and in brain stem areas that include precerebellar nuclei. *C*, single-molecule fluorescence *in situ* hybridization for miR-138-5p in the cerebellum (Left) and Hoechst nuclear staining (right). Layers of the cerebellar cortex are indicated (GCL, granule cell layer; ML, molecular layer; PCL, Purkinje cell layer; WM, white matter). Scale bar = 100 μm. *D*, the direct miR-138-5p target *Erbb4* (Daswani et al., 2022) is differentially expressed in inhibitory and excitatory neurons of the hippocampus, cerebellum, and precerebellar nuclei. *E*, mean expression of Erbb4 in neurons of the cerebellar cortex, with highest expression levels in granule cells. [Colour figure can be viewed at wileyonlinelibrary.com]

we analysed publicly available miRNA expression datasets (Rishik et al., 2025), which confirmed robust expression of miR-138-5p in the cerebellum (Fig. 1*B*). Additionally, we observed miR-138-5p expression in the pons and medulla oblongata (Fig. 1*B*), comprising regions containing precerebellar nuclei that are the origin of mossy fibres projecting to the cerebellum (Rodriguez & Dymecki, 2000). To investigate the localization of miR-138-5p within the adult cerebellar cortex, we performed fluorescence *in situ* hybridization of miR-138-5p. This analysis demonstrated strong expression of miR-138-5p in cerebellar GCs (Fig. 1*C*), which are the most abundant excitatory neurons in the cerebellar cortex and receive excitatory synaptic input from mossy fibres. Together, these results show that miR-138-5p is highly expressed in GCs of the adult cerebellar cortex. In addition, they suggest that miR-138-5p may be capable of modulating synaptic properties on both pre- and postsynaptic sides.

Erbb4 is a direct miR-138-5p target (Daswani et al., 2022) that has been implicated in pre- and postsynaptic function. We analysed publicly available single-cell RNA-seq datasets (Kozareva et al., 2021; Langlieb et al., 2023) to establish the expression levels of Erbb4 in the hippocampus, cerebellum and precerebellar nuclei. Our analysis revealed high expression levels of Erbb4 in inhibitory hippocampal interneurons and excitatory cerebellar GCs, whereas it is barely expressed in excitatory neurons of the hippocampus (Fig. 1*D* and *E*). At the synaptic level, Erbb4 has been associated with both postsynaptic and presynaptic compartments (Koopmans et al., 2019). Consistent with this, we observed robust Erbb4 expression in precerebellar nuclei located in the pons and medulla oblongata (Fig. 1*D*). These findings highlight prominent expression of Erbb4 in cerebellar GCs and presynaptic mossy fibres, and suggest that miR-138-5p could regulate synaptic properties on pre- and postsynaptic sides of cerebellar MF–GC synapses through modulating Erbb4.

### miR-138-5p negatively regulates synaptic strength at cerebellar MF–GC synapses

We generated mice with a conditional ROSA26 transgene ('miR-138 floxed'), which allows the expression of a sponge transcript that inactivates endogenous miR-138-5p upon Cre-recombinase expression (Daswani et al., 2022). Sponge transcripts sequester endogenous miRNA, thereby leading to miRNA inactivation and the de-repression of cognate target genes (Ebert & Sharp, 2010). We activated miR-138-sponge expression at the embryonic stage by crossing 138-floxed mice to the ubiquitous Cre-driver line CMV-Cre ('miR-138 sponge'). miR-138 floxed mice without CMV transgene served as controls. We confirmed expression of the 6×-miR-138 sponge in the cerebellar cortex of miR-138 sponge mice by lacZ staining (Fig. 2*A*).

To determine the role of miR-138-5p in excitatory neurotransmission within the cerebellum, we performed whole-cell patch clamp recordings from GCs in acute slices from adult mice. We isolated AMPAR-mediated synaptic transmission and recorded EPSCs from GCs of control (miR-138 floxed) and miR-138 sponge mice (Fig. 2*B*). Stimulation of single MFs resulted in reliable and fast AMPAR-mediated EPSCs in GCs (Fig. 2*C*). EPSC amplitudes were considerably larger in sponge than in control synapses (control: 91.1 pA, sponge: 124.7 pA; ∼37% increase) (Fig. 2*D* and *E*). We also noted a slight slowing of the EPSC decay (fast time constant: control, 1.25 ms; sponge: 1.56 ms; slow time constant: control, 12.87 ms; sponge, 18.43 ms) (Fig. 2*F* and *G*) without changes in the rise time (control: 0.333 ms, sponge: 0.318 ms) (Fig. 2*H*). In combination, the slower EPSC decay and the increased amplitude translate into a marked increase in EPSC charge by ∼44% in miR-138 sponge GCs (control: 0.275 pC, sponge: 0.398 pC) (Fig. 2*I*). These findings indicate that miR-138-5p negatively regulates synaptic efficacy at cerebellar MF–GC synapses by modulating AMPAR-mediated transmission.

Based on the enhanced synaptic strength at the cerebellar input stage of miR-138 sponge mice, we next recorded from Purkinje cells and investigated their excitatory parallel fibre input formed by GCs. Interestingly, spontaneous EPSCs recordings in Purkinje cells did not show any evidence for altered amplitudes (control: 11.3 pA; sponge: 10.3 pA) (Fig. 3*B* and *C*). Similarly, synaptic strength at parallel fibre to Purkinje cell synapses was largely unchanged in miR-138 sponge mice (EPSC input/output slope: control, 20.4 pA/mV *vs.* sponge, 12.7 pA/mV) (Fig. 3*D* and *E*). Synaptic facilitation assessed by PPRs at 50 ms was also similar in both genotypes (control: 1.83; sponge: 1.62) (Fig. 3*F*). Although a larger sample size may be required to rule out a role of miR-138 at parallel fibre synapses, we did not observe a pronounced increase in synaptic transmission as seen at MF–GC synapses (Fig. 2). These results thus point towards a synapse-specific regulatory role of this miRNA within the cerebellum.

### miR-138-5p negatively regulates the number of functional AMPARs in cerebellar GCs

The enhanced synaptic efficacy in MF–GC synapses observed following downregulation of miR-138-5p could be attributed to an increase in AMPAR function or number. To determine whether AMPARs are involved in the EPSC amplitude increase, we recorded spontaneous miniature EPSCs (mEPSCs) from cerebellar GCs (Fig. 4*A* and *B*). GCs from miR-138 sponge mice

exhibited increased mEPSC amplitudes (control: 13.2 pA, sponge: 15.5 pA) (Fig. 4*C*), indicating enhanced AMPAR-mediated transmission. This finding aligns with previous reports of reduced mEPSC amplitudes upon miR-138-5p overexpression in hippocampal cultures

(Siegel et al., 2009). Additionally, mEPSC charge was slightly increased in sponge GCs (control: 0.014 pC, sponge: 0.016 pC) (Fig. 4*D*), whereas the kinetics and frequency of mEPSCs were similar across both genotypes (rise time: control, 0.25 ms *vs*. sponge, 0.22 ms;

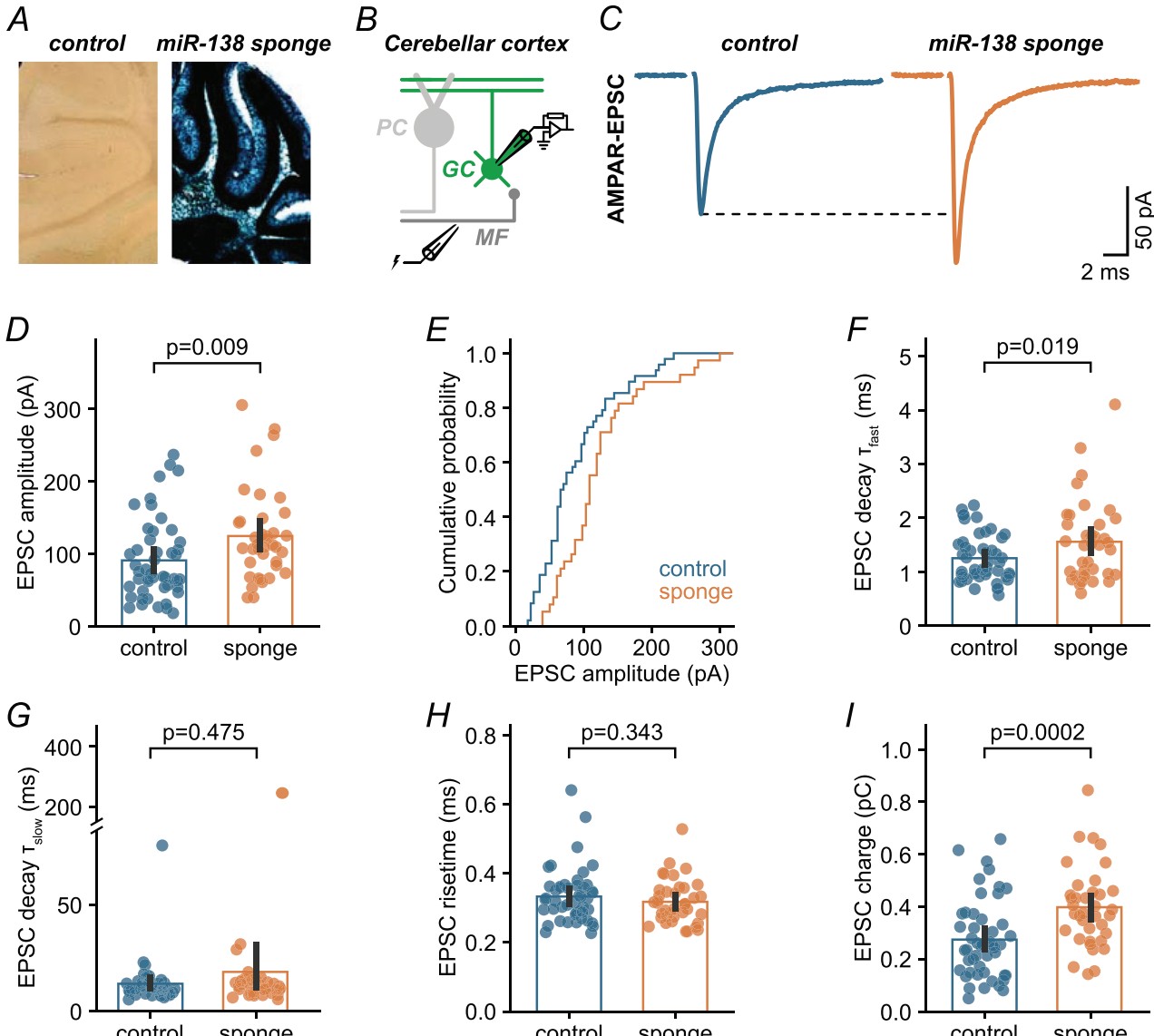

**Figure 2. miR-138-5p negatively regulates synaptic strength at cerebellar MF–GC synapses**
*A*, left: lacZ staining of the cerebellum of a control mouse without CMV transgene ('miR-138 floxed', see Methods). Right: lacZ staining of a miR-138 sponge mouse, confirming expression of the 6×-miR-138 sponge in the cerebellar cortex. *B*, cartoon illustrating MF–GC recordings in acute cerebellar slices from adult control and miR-138 sponge mice. *C*, left: example average EPSC recorded from a control MF–GC synapse (average of 24 sweeps). Right: example EPSC from a miR-138 sponge synapse (average of 18 sweeps). Stimulation artifacts are blanked. *D*, quantification of EPSC amplitude for control and sponge conditions [Cohen's *d*: 0.56; 95% confidence interval (CI) = 0.13–0.99; control: *n* = 48 cells, *N* = 5 mice; sponge: *n* = 38 cells, *N* = 4 mice]. *E*, cumulative probability of mean EPSC amplitudes for control and sponge. *F*, the fast EPSC decay time constant is slightly slower in miR-138 sponge GCs (Cohen's *d*: 0.51; 95% CI = 0.08–0.90). *G*, quantification of the slow EPSC decay time constant in both genotypes (Cohen's *d*: 0.21; 95% CI = −0.24 to 0.59). *H*, EPSC 10%–90% rise time is similar for both genotypes (Cohen's *d*: −0.21; 95% CI = −0.59 to 0.22). *I*, miR-138 sponge MF–GC EPSCs have increased charge transfer (Cohen's *d*: 0.82; 95% CI = 0.37–1.28). Bars are means with error bars representing the 95% CI. [Colour figure can be viewed at wileyonlinelibrary.com]

decay: control, 0.86 ms *vs.* sponge, 0.81 ms; frequency: control, 0.27 Hz *vs.* sponge, 0.37 Hz) (Fig. 4*E*–*G*). mEPSC amplitude mainly depends on synaptic AMPAR numbers and conductance. To determine whether miR-138-5p influences the AMPAR single-channel conductance or the number of functional receptors, we performed

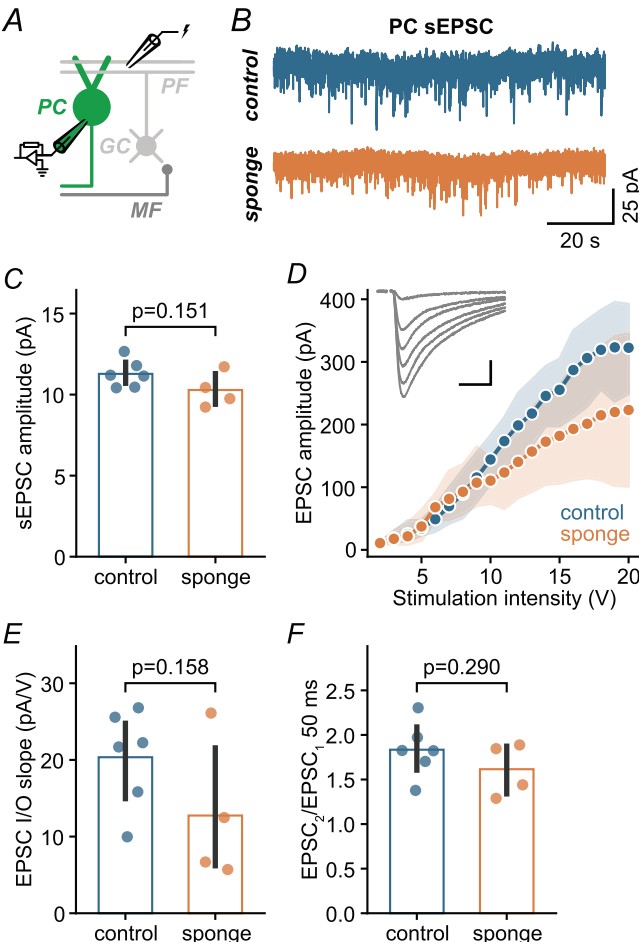

**Figure 3. miR-138-5p sponge expression does not increase synaptic efficacy at PF–PC synapses**

*A*, cartoon illustrating recordings from control and sponge Purkinje cells (PCs). *B*, representative examples of spontaneous EPSC recordings in control and sponge Purkinje cells. *C*, quantification of spontaneous EPSC amplitudes for Purkinje cells of both genotypes. sEPSCs were not increased in miR-138 sponge mice (Cohen's *d*: −1.06; 95% CI = −2.63 to 0.84; control: *n* = 6 cells, *N* = 5 mice; sponge: *n* = 4 cells, *N* = 4 mice). *D*, EPSC amplitude *vs.* stimulation intensity for control and sponge parallel fibre to Purkinje cell (PF–PC) synapses. Dots are averages of ten EPSCs, shaded areas represent the 95% CI. Inset shows representative EPSCs at six stimulation intensities (scale bars = 10 ms and 100 pA). *E*, slopes of PF–PC EPSC input/output relationship obtained by linear regression (Cohen's *d*: −1.00; 95% CI = −3.26 to 1.34; control: *n* = 6 cells, *N* = 5 mice; sponge: *n* = 4 cells, *N* = 4 mice). *F*, paired-pulse ratio (EPSC$_2$/EPSC$_1$) at 20 Hz stimulation frequency was similar in control and sponge PF–PC synapses (Cohen's *d*: −0.72; 95% CI = −2.16 to 1.34). Bars are means with error bars representing the 95% CI. [Colour figure can be viewed at wileyonlinelibrary.com]

peak-scaled non-stationary fluctuation analysis of the mEPSC decay (Traynelis et al., 1993) (Fig. 4*H* and *I*). Single-channel conductance was comparable between conditions (control: 14.1 pS, sponge: 14.0 pS) (Fig. 4*J*), but the number of channels was greater in sponge GCs (control: 9.8, sponge: 14.7) (Fig. 4*K*). This suggests that miR-138-5p negatively regulates the number of synaptic AMPARs that are open at the peak of the synaptic current in cerebellar GCs. Together, mEPSC analysis revealed larger mEPSC amplitudes in miR-138 sponge GCs, consistent with the idea that miR-138 negatively regulates the number of functional AMPARs.

### miR-138-5p suppresses presynaptic function

Interestingly, the mEPSC amplitude increase following miR-138 sponge expression only accounts for about half of the overall increase in synaptic efficacy upon presynaptic AP stimulation (17.0% *vs.* 36.9%), suggesting that the role of miR-138-5p at MF–GC synapses extends beyond simply regulating postsynaptic AMPAR levels. To determine whether presynaptic mechanisms contribute to the enhanced synaptic efficacy at miR-138 sponge MF–GC synapses, we analysed quantal content (a proxy for neurotransmitter release) derived from AP-evoked EPSCs and mEPSC recordings in the same cells. We found that quantal content was increased in miR-138 sponge GCs (control: 8.3, sponge: 10.56) (Fig. 5*A*), indicating enhanced presynaptic glutamate release from MFs. Further analysis of evoked EPSC variance revealed a decreased coefficient of variation (CV), which is inversely correlated with presynaptic release probability, following downregulation of miR-138-5p (control: 0.24, sponge: 0.17) (Fig. 5*B* and *C*), suggesting an increased release probability. To further examine changes in release probability, we recorded EPSC paired-pulse ratios (PPRs) (Fig. 5*D*). MF–GC synapses show a combination of synaptic facilitation and depression upon paired-pulse stimulation, with depression prevailing at shorter intervals (Saviane & Silver, 2006). PPRs were reduced in miR-138 sponge GCs (10 ms: control, 0.74 *vs.* sponge, 0.59; 5 ms: control, 0.71 *vs.* sponge, 0.52) (Fig. 5*E* and *F*), consistent with an increased release probability. A reduction in PPR could also arise from altered receptor desensitization. To test this possibility, we fit the relationship between PPR and inter-stimulus interval with bi-exponential fits (Fig. 5*G*). The fast time constants were comparable for both conditions (control: 11.1 ms, sponge: 11.7 ms), suggesting a similar time course of AMPAR desensitization (DiGregorio et al., 2007; Saviane & Silver, 2006), therefore further supporting an increase in release probability in miR-138 sponge MF–GC synapses. PPRs of individual connections were negatively correlated with synaptic strength at control synapses (*R* =

−0.58) (Fig. 5*H*). Interestingly, we only rarely observed facilitating MF–GC synapses in miR-138 sponge mice (cells with PPR >1 at any interval: sponge, 5.7%; control, 42.6%), resulting in a weaker correlation between PPR and EPSC amplitude (*R* = −0.22) (Fig. 5*I*). In summary, miR-138 sponge synapses exhibit increased glutamate release and reduced PPR, indicating that miR-138-5p negatively regulates presynaptic release probability.

## miR-138-5p counteracts short-term depression at MF–GC synapses

To further analyse the effect of miR-138-5p down-regulation on release probability and synaptic short-term plasticity, we performed high-frequency train stimulation experiments. MF–GC synapses are capable of operating with a high bandwidth of presynaptic AP frequencies (Delvendahl & Hallermann, 2016) and typically exhibit

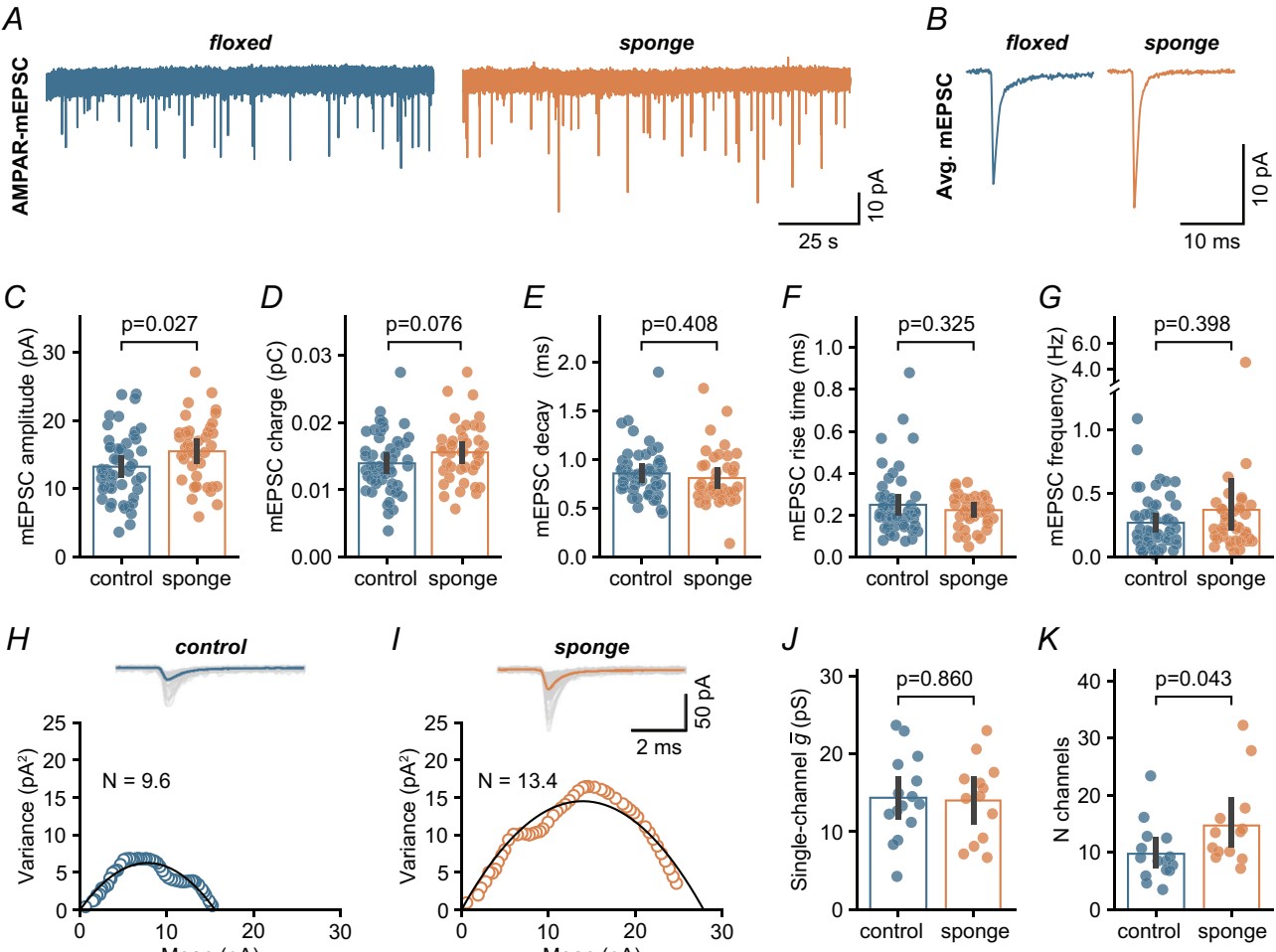

**Figure 4. miR-138-5p negatively regulates the number of functional AMPARs in cerebellar GCs**
*A*, left: example mEPSC recording from a control GC. Right: example recording from a miR-138 sponge GC. *B*, average mEPSC waveforms of the examples shown in (*A*). *C*, mean mEPSC amplitude is bigger in sponge than in control GCs (Cohen's *d*: 0.48; 95% CI = 0.06–0.90; control: *n* = 53 cells, *N* = 5 mice; sponge: *n* = 41 cells, *N* = 4 mice). *D*, quantification of mEPSC charge for both conditions (Cohen's *d*: 0.37; 95% CI = −0.04 to 0.78). *E*, quantification of mEPSC decay time constant, calculated from exponential fits to the average mEPSC waveform of each cell (Cohen's *d*: −0.18; 95% CI = −0.59 to 0.25). *F*, mEPSC rise time for control and sponge groups (Cohen's *d*: −0.20; 95% CI = −0.54 to 0.16). *G*, the frequency of mEPSCs is similar in both genotypes (Cohen's *d*: 0.21; 95% CI = −0.21 to 0.56). *H–I*, example mEPSC variance-mean analysis. Shown are representative individual mEPSCs overlaid with average (top) and peak-scaled mEPSC decay variance *vs.* mean amplitude for both genotypes (bottom). Number of AMPARs (*N*) is indicated. *J*, AMPAR single-channel conductance was unaltered by miR-138 sponge expression (Cohen's *d*: −0.07; 95% CI = −0.86 to 0.73; control: *n* = 15 cells, *N* = 5 mice; sponge: *n* = 13 cells, *N* = 4 mice). *K*, the number of open AMPARs is higher in sponge GCs (Cohen's *d*: 0.79; 95% CI = −0.08 to 1.44). Bars are means with error bars representing the 95% CI. [Colour figure can be viewed at wileyonlinelibrary.com]

prominent short-term depression upon sustained stimulation (Saviane & Silver, 2006). We recorded short trains of EPSCs at 300 Hz (Fig. 6*A*), followed by single pulses to monitor recovery from depression (Hallermann et al., 2010). In miR-138 sponge GCs, we observed increased synaptic depression (Fig. 6*B* and *C*), evident from a faster time constant of depression (control: 6.0 ms, sponge: 4.0 ms) (Fig. 6*D*) and reduced

steady-state amplitudes (control: 0.21, sponge: 0.14) (Fig. 6*E*). This more pronounced depression indicates an increase in release probability. Nevertheless, the overall larger EPSCs in miR-138 sponge mice caused an increase in phasic charge transfer (see Methods) during high-frequency train stimulation (control: 0.27 pC, sponge: 0.34 pC) (Fig. 6*F*). To estimate the size of the readily releasable pool (RRP) of synaptic vesicles and

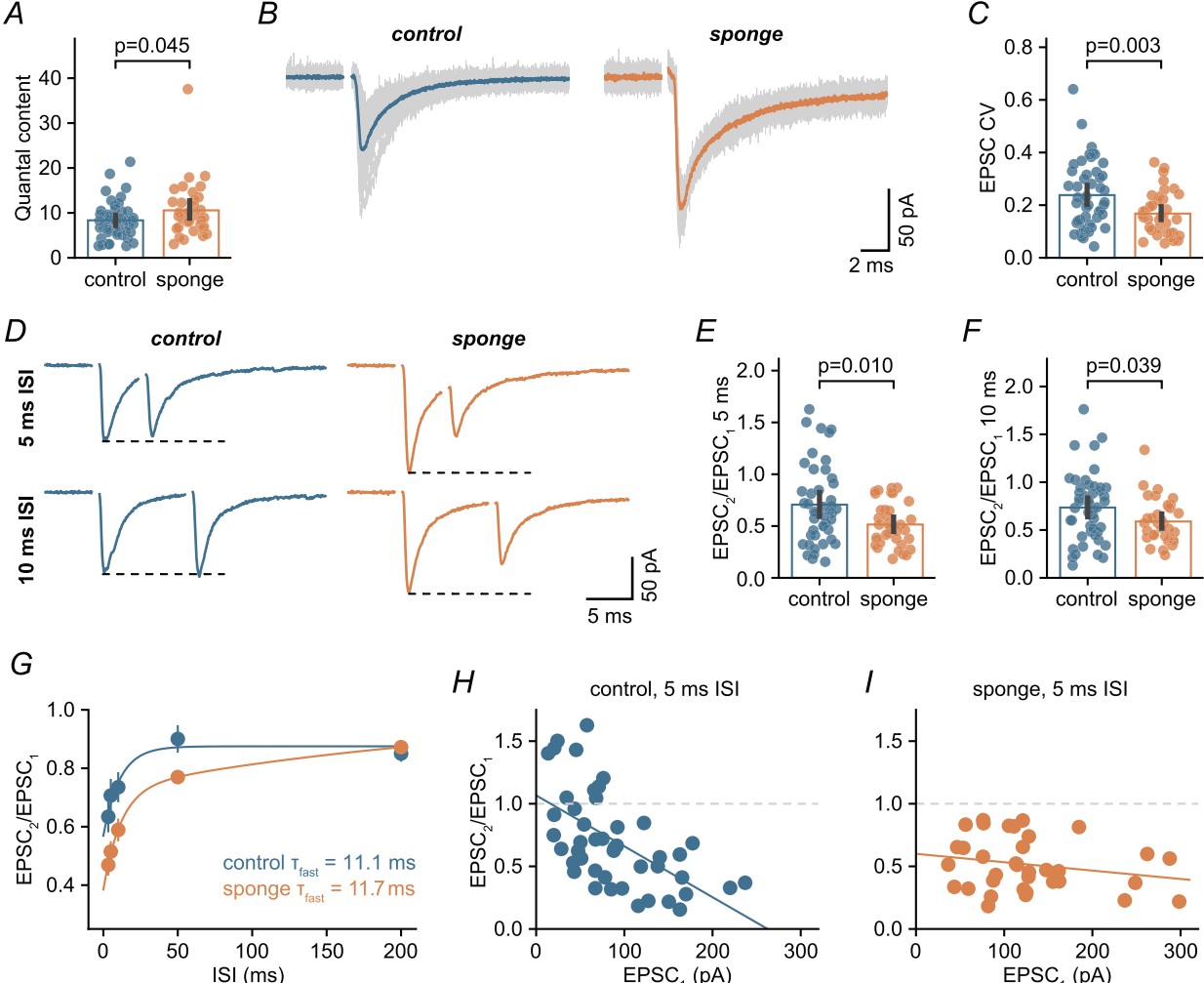

**Figure 5. miR-138-5p suppresses presynaptic function**
*A*, quantification of quantal content (= EPSC/mEPSC) for both genotypes (Cohen's *d*: 0.44; 95% CI = −0.01 to 0.83; control: *n* = 46 cells, *N* = 5 mice; sponge: *n* = 36 cells, *N* = 4 mice). *B*, left: example individual EPSCs (grey) overlaid with the average (blue) for a control MF–GC synapse. Right: example EPSCs from a sponge synapse overlaid with average (orange). Stimulation artifacts are blanked. *C*, coefficient of variation (CV) of evoked EPSC amplitudes is decreased in miR-138 sponge GCs (Cohen's *d*: −0.65; 95% CI = −1.02 to −0.23; control: *n* = 48 cells, *N* = 5 mice; sponge: *n* = 38 cells, *N* = 4 mice). *D*, left: example paired-pulse responses from a control cerebellar GC (average of 5 consecutive sweeps). Stimulation artifacts are blanked. Right: example recording from a sponge GC. *E*, paired-pulse ratios (PPRs, *i.e.* EPSC$_2$/EPSC$_1$) at 200 Hz for control and sponge GCs (Cohen's *d*: −0.60; 95% CI = −0.98 to −0.20; control: *n* = 45 cells, *N* = 5 mice; sponge: *n* = 35 cells, *N* = 4 mice). *F*, same as in (*E*), but for 100 Hz (Cohen's *d*: −0.48; 95% CI = −0.88 to −0.04; control: *n* = 47 cells, *N* = 5 mice; sponge: *n* = 35 cells, *N* = 4 mice). Bars are means with error bars representing the 95% CI. *G*, PPR *vs.* inter-stimulus interval (ISI). Dots are averages of 29–47 cells per ISI; error bars represent the SEM. Lines are bi-exponential fits, time constant of the fast component is indicated. *H*, EPSC PPR *vs.* amplitude of the first EPSC in GCs of control mice for 200 Hz stimulation with linear regression. *I*, same as in (*H*), but for miR-138 sponge GCs. [Colour figure can be viewed at wileyonlinelibrary.com]

apparent release probability, we analysed cumulative EPSC amplitudes (Schneggenburger et al., 1999) (Fig. 6G). The steady-state recruitment slope was comparable between both genotypes (control: 4.6 pA/ms; sponge: 5.0 pA/ms) (Fig. 6H). RRP size was similar between control and miR-138 sponge synapses (control: 14.6, sponge: 16.5) (Fig. 6I), whereas release probability was slightly increased in miR-138 sponge animals (control:

0.56, sponge: 0.63; Cohen's *d*: 0.44) (Fig. 6J). These EPSC high-frequency train recordings demonstrate altered synaptic short-term dynamics, with a stronger degree of depression in miR-138 sponge mice. Collectively, the results from paired-pulse EPSCs and train recordings support the notion that miR-138-5p negatively regulates release probability and counteracts short-term depression at cerebellar MF–GC synapses.

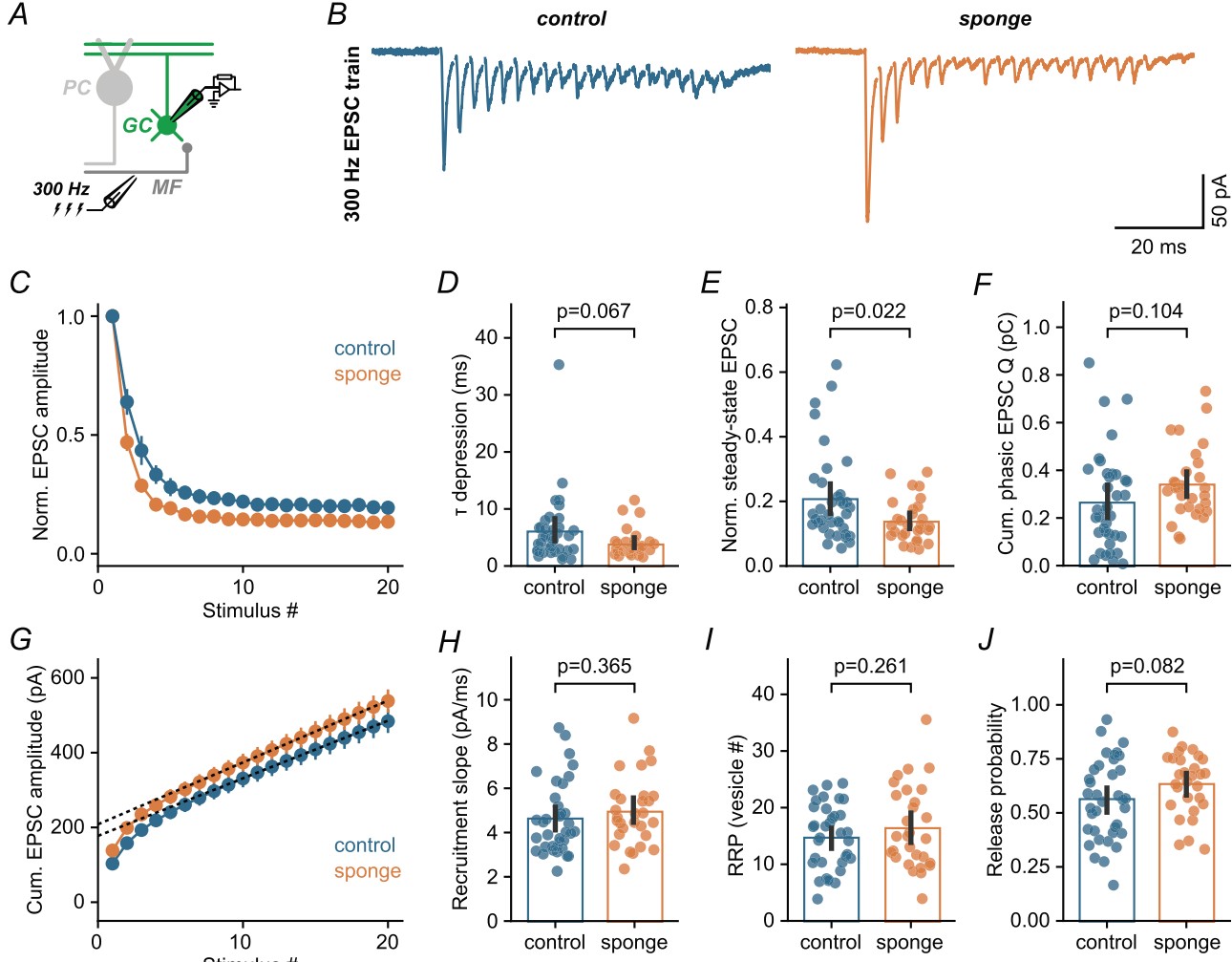

**Figure 6. miR-138-5p counteracts short-term depression at MF–GC synapses**
*A*, schematic of 300 Hz train recording from cerebellar GCs. *B*, left: example EPSC responses upon 300 Hz stimulation for a control GC. Average of five sweeps; stimulation artifacts are blanked for clarity. Right: example 300 Hz train for a miR-138 sponge synapse. *C*, normalized EPSC amplitude for both genotypes (control: *n* = 37 cells, *N* = 5 mice; sponge: *n* = 29 cells, *N* = 4 mice). Error bars are the SEM. *D*, time constant of synaptic depression for control and sponge synapses (Cohen's *d*: −0.43; 95% CI = −0.74 to 0.04). *E*, miR-138 sponge expression decreases the normalized steady-state EPSC amplitude (Cohen's *d*: −0.58; 95% CI = −0.92 to −0.17). *F*, cumulative phasic EPSC charge for both genotypes (Cohen's *d*: 0.41; 95% CI = −0.10 to 0.93). *G*, cumulative EPSC amplitude for 300 Hz train stimulation. Error bars represent the SEM; dashed lines are linear fits to the last 10 datapoints (control: *n* = 37 cells, *N* = 5 mice; sponge: *n* = 29 cells, *N* = 4 mice). *H*, EPSC recruitment slope for both genotypes, obtained by linear regression of the cumulative EPSC amplitude at steady state (Cohen's *d*: 0.22; 95% CI = −0.29 to 0.70). *I*, readily releasable pool (RRP) of synaptic vesicles, calculated from cumulative EPSC analysis (Cohen's *d*: 0.28; 95% CI = −0.22 to 0.80). *J*, Apparent release probability (first EPSC/cumulative EPSC) is higher in miR-138 sponge MF–GC synapses (Cohen's *d*: 0.44; 95% CI = −0.07 to 0.94). Bars are means with error bars representing the 95% CI. [Colour figure can be viewed at wileyonlinelibrary.com]

## miR-138-5p negatively regulates presynaptic release probability

To determine whether increased release probability accounts for the observed short-term depression effects in miR-138 sponge GCs, we applied a synaptic short-term plasticity model (Tsodyks et al., 1998) to our experimental data (Fig. 7*A* and *B*). Across recordings, the fitted model parameter 'utilization of synaptic efficacy' ($U_{SE}$), which corresponds to release probability, was elevated in the sponge condition (control: 0.34, sponge: 0.45) (Fig. 7*C*). These computational modelling results are consistent with a higher release probability driving the more pronounced short-term depression upon high-frequency stimulation in miR-138 sponge expressing synapses. In addition, the fitted model indicated a faster

recovery time constant for the miR-138 sponge condition (control: 97.1 ms, sponge: 60.8 ms) (Fig. 7*D*), pointing towards faster recovery from depression. We further analysed the recovery from synaptic depression following 300 Hz stimulation in control and sponge GCs (Fig. 7*E*). Recovery at MF–GC synapses follows a bi-exponential time course (Hallermann et al., 2010). Consistent with the modelling results, the fast component of EPSC recovery was enhanced in sponge GCs, with a faster time constant and larger fractional amplitude (tau: control, 155.8 ms *vs.* sponge, 96.9 ms; amplitude: control, 0.52 *vs.* sponge, 0.63) (Fig. 7*F* and *G*). Thus, the increased synaptic depression in miR-138 sponge synapses is accompanied by faster recovery. Together, these findings provide further evidence that miR-138-5p suppresses release probability at MF–GC synapses.

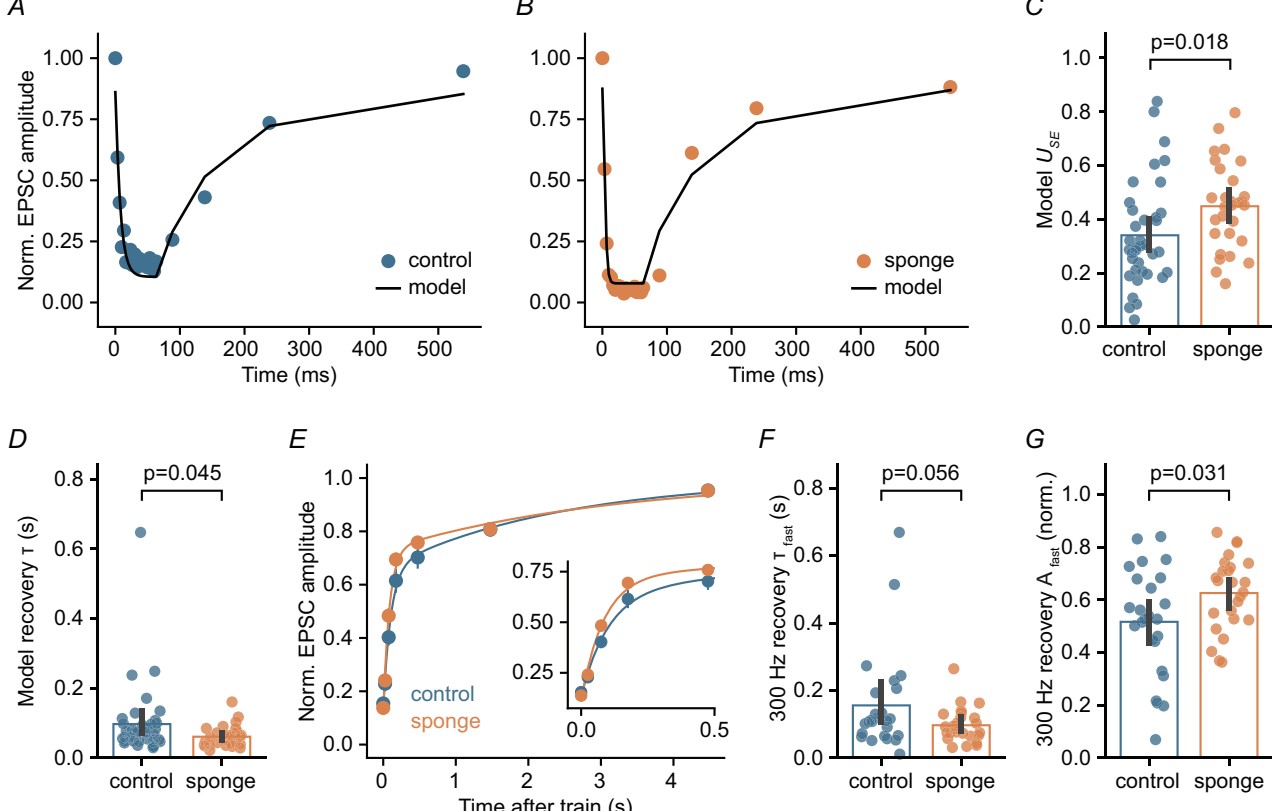

**Figure 7. miR-138-5p negatively regulates presynaptic release probability**
*A*, representative normalized EPSC amplitudes for 300 Hz train stimulation and recovery pulses. Black line represents a Tsodyks–Markram short-term plasticity model fit to the experimental data. *B*, same as in (*A*), but for an example miR-138 sponge synapse. *C*, quantification of model parameter $U_{SE}$ (Cohen's *d*: 0.60; 95% CI = 0.07–1.13). *D*, the time constant of recovery from depression is faster in models fitted to sponge data from high-frequency train stimulation (Cohen's *d*: –0.44; 95% CI = –0.68 to 0.01). *E*, normalized EPSC amplitude *vs.* time after 300 Hz trains for both conditions. Lines are bi-exponential fits to the recovery time course. Inset shows initial phase of recovery on a shorter time scale. *F*, the fast time constant of recovery from depression is shorter at miR-138 sponge synapses (Cohen's *d*: –0.53; 95% CI = –0.89 to –0.01; control: *n* = 25 cells, *N* = 5 mice; sponge: *n* = 25 cells, *N* = 4 mice). *G*, quantification of the relative amplitude of the fast recovery component for both genotypes (Cohen's *d*: 0.62; 95% CI = 0.04–1.17). Bars are means with error bars representing the 95% CI. [Colour figure can be viewed at wileyonlinelibrary.com]

## Discussion

In the present study, we have revealed a general repressive mechanism for excitatory synaptic strength mediated by miR-138-5p at excitatory mossy fibre to granule cell synapses in the mammalian cerebellum. The expression of a miR-138-5p sponge construct caused an increase in postsynaptic mEPSC size via an increase in functional AMPAR numbers at synapses. In addition, sequestering of miR-138-5p led to enhanced glutamate release, driven by an increase in presynaptic release probability. miR-138-5p thus suppresses synaptic strength at excitatory cerebellar synapses through distinct presynaptic and postsynaptic mechanisms. This dual regulatory role underscores the importance of miR-138-5p in modulating synaptic excitation at the cerebellar input layer.

The distinct modulations at both sides of the synapse indicate that the effect of miR-138-5p on excitatory synaptic transmission extends beyond a simple regulation of an individual target mRNA. miR-138-5p rather acts as a general repressor of synaptic strength. Many different proteins influence the functional properties of individual synapses (Cizeron et al., 2020; Van Oostrum et al., 2023; Zhu et al., 2018). Loss or downregulation of a given synaptic protein often causes a disruption of synaptic function, suggesting important roles of individual proteins for promoting synaptic transmission. Our data provide evidence for a general, negative regulation of excitatory synaptic strength at cerebellar input synapses. A mechanism that suppresses synaptic strength under baseline conditions could enable dynamic modulation of synaptic function, such as activity-dependent (Mapelli et al., 2015) or homeostatic (Delvendahl & Müller, 2019) plasticity. A miRNA-dependent mechanism controlling synaptic strength is thus an intriguing candidate for adaptive plasticity of synaptic function. This hypothesis is supported by studies linking miR-138 to memory processes in rodents (Li et al., 2018) and humans (Schröder et al., 2014).

Our data point towards a brain region-specific (and potentially synapse type-specific) role of miR-138-5p for synaptic transmission. In the hippocampus, inactivation of miR-138-5p selectively affects inhibitory transmission onto pyramidal neurons (Daswani et al., 2022). In the cerebellum, we identified a strong impact of miR-138 sponge expression on excitatory MF–GC synapses. The negative regulation of excitatory synaptic strength may be specific to MF–GC synapses because transmission at parallel fibre synapses onto Purkinje cells, which also express miR-138-5p (Zolboot et al., 2025), was not increased. It will be interesting to further elucidate the function of miR-138-5p in Purkinje cells, and to investigate whether this miRNA has effects on inhibitory synapses in the cerebellum. A presumably synapse type-specific regulatory effect suggests that miRNAs like miR-138-5p could fine-tune synaptic properties in a circuit-specific manner.

Overexpression of miR-138-5p decreased mEPSC amplitudes in a hippocampal culture preparation (Siegel et al., 2009). In combination with our observation of enhanced mEPSC amplitudes upon miR-138 sponge expression in a cerebellar slice preparation, these findings strongly suggest that miR-138-5p negatively regulates the number of functional AMPARs. Our analysis revealed an increase in the number of receptors that open at the peak of the synaptic current, which may be the result of an enhanced number or open probability of synaptic AMPARs. miR-138 could modulate AMPAR function by repressing AMPAR translation (Hanley, 2021) or by altering receptor gating properties through an influence on AMPAR complex composition (Schwenk et al., 2012).

A previous study identified Erbb4 as a direct target of miR-138-5p (Daswani et al., 2022). Erbb4 is a receptor tyrosine kinase that can interact with postsynaptic density proteins (Garcia et al., 2000). Erbb4 has a prominent role in inhibitory hippocampal interneurons, where miR-138 sponge expression caused an increase in inhibitory miniature event frequency (Daswani et al., 2022). The excitatory GCs in the cerebellar cortex also express Erbb4, implying a region-specific role of this receptor tyrosine kinase. Interestingly, Erbb4 was shown to regulate translation of several synaptic proteins, including the pore-forming AMPAR subunit GluA4 (Bernard et al., 2022). In the cerebellum, Erbb4 co-precipitates with GluA4 (Pelkey et al., 2015), which is the predominant AMPAR subunit in GCs (Kita et al., 2021). In the present study, inactivation of miR-138-5p increased mEPSC amplitudes in cerebellar GCs. This increase is probably mediated by increasing synaptic AMPAR numbers, which would be consistent with miR-138-5p targeting (and thus repressing) Erbb4, which in turn promotes GluA4 translation. A regulation of AMPAR subunits by miRNAs has been demonstrated for GluA1 and GluA2. miR-183/96 double knockout increased GluA1 levels at the calyx of Held synapse (Krohs et al., 2021) and GluA1 is also a direct target of miR-137 (Olde Loohuis et al., 2015), as well as miR-92a (Letellier et al., 2014). Likewise, miR-186 (Silva et al., 2019), miR-124 (Ho et al., 2014) and miR-181 (Saba et al., 2012) regulate GluA2 levels in hippocampal cultures. miRNA-dependent regulation of AMPAR levels may thus be a pervasive mechanism of controlling excitatory synaptic strength (Hanley, 2021).

In addition to postsynaptic effects on the level of AMPARs, miR-138-5p also prominently affected presynaptic function at cerebellar MF–GC synapses. The combined presynaptic and postsynaptic effect on synaptic efficacy is reminiscent of results on another miRNA complex (miR-183/96) at the calyx of Held synapse in the auditory brainstem (Krohs et al., 2021). In contrast to our observation of a miR-138-5p-dependent regulation of

release probability at cerebellar MF–GC synapses, global knockout of miR-183 and miR-96 increased RRP size without affecting release probability at the calyx of Held. Nonetheless, the convergent negative regulation of both synaptic compartments by distinct miRNAs in different brain regions points towards a general mechanism for controlling excitatory synaptic strength. This notion is further supported by data in hippocampal cultures, where miR-485 regulates levels of specific pre- and post-synaptic proteins (Cohen et al., 2011), as well as from the *Drosophila* neuromuscular junction, where miR-34 controls synaptogenesis by regulating distinct pre- and postsynaptic genes (McNeill et al., 2020).

Although our data suggest a modulation of synaptic strength by miR-138 via a regulation of specific mRNAs, we cannot exclude the possibility that developmental effects contribute to the synaptic phenotype in sponge animals. There were no gross morphological differences between controls and sponge cerebella, but we have not addressed, for example, synaptic morphology. In the future, temporally controlled or cell-type specific down-regulation of miR-138 could provide more insights into this question.

miRNAs can potentially influence release probability by repressing the translation of presynaptic proteins that are involved in setting release probability. Although miR-138-5p has some predicted targets with presynaptic localization, experimental evidence for its regulation of presynaptic targets is currently lacking. The direct miR-138-5p target Erbb4 (Daswani et al., 2022) is linked to both presynaptic and postsynaptic compartments (Koopmans et al., 2019) and is expressed in precerebellar nuclei from where mossy fibres originate. Intriguingly, Erbb4 may also promote presynaptic release probability (Wang et al., 2018) and could thus be involved in the altered release properties that we observed in miR-138 sponge mice. Nevertheless, it remains unclear whether Erbb4 is also a direct target of miR-138-5p in the cerebellum, and also whether the effects of miR-138-5p inactivation on presynaptic function are a result of the expression of this miRNA in presynaptic cells or trans-synaptic signalling. The negative regulation of pre-synaptic function by miR-138-5p is also consistent with findings in *Drosophila*, where another miRNA, miR-130, was found to suppress quantal content via negative regulation of the active zone protein Bruchpilot and $Ca^{2+}$ influx (Tsurudome et al., 2010). A negative regulation of active zone proteins was also shown for miR-153 and miR-137 in the mouse hippocampus (Mathew et al., 2016; Siegert et al., 2015). miRNAs may thus control the levels of key active zone proteins, thereby regulating presynaptic functions such as release probability.

The influence of miR-138-5p on presynaptic release appears to promote low release probability synapses with facilitating short-term dynamics. This effectively broadens the distribution of synaptic strengths and short-term plasticity behaviour across the population of MF–GC synapses, which may have important implications for cerebellar network function. For example, diversity of synaptic short-term dynamics at the cerebellar input layer could serve as a mechanism for temporal learning (Barri et al., 2022). A broad distribution of MF–GC short-term plasticity may also enhance pattern separation (Chabrol et al., 2015), which is considered to be a key computation of the cerebellar cortex. Our data indicate that miR-138-5p contributes to synaptic diversity at the cerebellar input layer and might support cerebellar computations by endowing a fraction of MF–GC synapses with facilitating properties. It will be highly interesting to investigate whether inactivation of miR-138-5p affects cerebellum-dependent behaviour or learning.

In summary, we have investigated the role of miR-138-5p for excitatory synaptic transmission in the mouse cerebellum. Expressing an inactivating sponge construct of this miRNA prominently increased synaptic efficacy at MF–GC synapses through combined pre-synaptic and postsynaptic mechanisms. We conclude that miR-138-5p acts as a negative regulator of synaptic strength at the cerebellar input layer. The general repression of synaptic efficacy across synaptic compartments provides a powerful regulatory mechanism for synaptic function, which may be crucial for maintaining proper cerebellar network function.

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

## Additional information

### Data availability statement

All data supporting the results in the manuscript are included within the figures and/or text. Raw data are available from the corresponding author upon reasonable request.

### Competing interests

The authors declare that they have no competing interests.

## Author contributions

I.D., J.W., G.S. and M.M. were responsible for conceptualization. I.D., R.D., P-.L.G. and N.U. were responsible for investigations. I.D., R.D., J.W., P-.L.G. and N.U. were responsible for formal analysis. I.D. was responsible for visualization. I.D. was responsible for writing the original draft. I.D., G.S. and M.M. were responsible for funding acquisition. J.W., P-.L.G., G.S. and M.M. were responsible for reviewing and editing. G.S. and M.M. were responsible for supervision

## Funding

This work was supported by the Swiss National Science Foundation (grants PZ00P3_174 018 to ID and PP00P3_144 816 to MM), the German Research Foundation (grant no. 535 029 399 to ID), a European Research Council Starting Grant (SynDegrade; grant 679 881 to MM) and a University Research Priority Program 'Adaptive Brain Circuits in Development and Learning' grant by the University of Zurich (to MM).

## Acknowledgements

We thank Yannick Ruhl and Jochen Schwenk for critically reading the manuscript.

## Supporting information

Additional supporting information can be found online in the Supporting Information section at the end of the HTML view of the article. Supporting information files available:

**Peer Review History**

