## [Peer Review History · The Journal of Physiology]

microRNA-138-5p suppresses excitatory synaptic strength at the cerebellar input layer

Igor Delvendahl, Reetu Daswani, Jochen Winterer, Pierre-Luc Germain, Nora Maria Uhr, Gerhard Schratt, and Martin Mueller

DOI: 10.1113/JP288019

Corresponding author(s): Igor Delvendahl (igor.delvendahl@physiologie.uni-freiburg.de)

The following individual(s) involved in review of this submission have agreed to reveal their identity: Zoltan Nusser (Referee #1)

Review Timeline:

Submission Date:	31-Oct-2024
Editorial Decision:	09-Dec-2024
Revision Received:	25-Mar-2025
Accepted:	14-Apr-2025

Senior Editor: Katalin Toth

Reviewing Editor: Samuel Young

Transaction Report:

Dear Dr Delvendahl,

Re: JP-RP-2024-288019 "microRNA-138-5p suppresses excitatory synaptic strength at the cerebellar input layer" by Igor Delvendahl, Reetu Daswani, Jochen Winterer, Pierre-Luc Germain, Nora Maria Uhr, Gerhard Schrott, and Martin Mueller

Thank you for submitting your manuscript to The Journal of Physiology. It has been assessed by a Reviewing Editor and by 2 expert referees and we are pleased to tell you that it is potentially acceptable for publication following satisfactory major revision.

REVISION CHECKLIST:

Please upload two versions of your manuscript text: one with all relevant changes highlighted and one clean version with no

changes tracked. The manuscript file should include all tables and figure legends, but each figure/graph should be uploaded as separate, high-resolution files.

We look forward to receiving your revised submission.

Yours sincerely,

Katalin Toth
Senior Editor
The Journal of Physiology

REQUIRED ITEMS

- Author photo and profile. First or joint first authors are asked to provide a short biography (no more than 100 words for one author or 150 words in total for joint first authors) and a portrait photograph. These should be uploaded and clearly labelled together in a Word document with the revised version of the manuscript. See Information for Authors for further details.

- You must start the Methods section with a paragraph headed Ethical approval (https://jp.msubmit.net/cgi-bin/main.plex?form_type=display_requirements#methods).

Research must comply with The Journal's policies regarding animal experiments (<https://physoc.onlinelibrary.wiley.com/hub/animal-experiments>) and adherence to these policies must be stated in the manuscript.

Authors should confirm in their Methods section that their experiments were carried out according to the guidelines laid down by their institution's animal welfare committee, including an ethics approval reference number. The Methods section must contain a statement about access to food, water and housing, details of the anaesthetic regime: anaesthetic used, dose and route of administration, and method of killing the experimental animals.

- Please upload separate high-quality figure files via the submission form.

- Please ensure that the Article File you upload is a Word file.

- Your paper contains Supporting Information of a type that we no longer publish, including supplementary tables and figures. Any information essential to an understanding of the paper must be included as part of the main manuscript and figures. The only Supporting Information that we publish are video and audio, 3D structures, program codes and large data files. Your revised paper will be returned to you if it does not adhere to our Supporting Information Guidelines.

- Please include an Abstract Figure file, as well as the Figure Legend text within the main article file. The Abstract Figure is a piece of artwork designed to give readers an immediate understanding of the research and should summarise the main conclusions. If possible, the image should be easily 'readable' from left to right or top to bottom. It should show the physiological relevance of the manuscript so readers can assess the importance and content of its findings. Abstract Figures should not merely recapitulate other figures in the manuscript. Please try to keep the diagram as simple as possible and without superfluous information that may distract from the main conclusion(s). Abstract Figures must be provided by authors no later than the revised manuscript stage and should be uploaded as a separate file during online submission labelled as File Type 'Abstract Figure'. Please also ensure that you include the figure legend in the main article file. All Abstract Figures

should be created using BioRender. Authors should use The Journal's premium BioRender account to export high-resolution images. Details on how to use and access the premium account are included as part of this email.

EDITOR COMMENTS

Reviewing Editor:

This manuscript examines the role of micro-RNA, miR-138-5p, in regulating excitatory synaptic transmission at the mossy-fiber to granule synapses in the cerebellum. Both reviewers found the data that as highly rigorous and support the conclusions drawn. In addition, both felt that the data added to our understanding of synaptic transmission in the cerebellum. However, the reviewers had some concerns. To address the concern of whether the presynaptic phenotypes are observed are specific to the presynaptic cell or a postsynaptic change via retrograde signal, the authors should provide data on miR138-5p expression in the brainstem nuclei, or do selective expression of the miR138-5p in granule cells. In addition, there is concern about phenotypes observed may be exaggerated by a single mutant animal in the data set. The authors need to address this concern. The authors need to carefully revise and rewrite their manuscript give the positive and careful comments by the reviewers. In addition, the authors need to move all supplemental data into the main text to comply with Journal policy.

Please also see 'Required Items' above.

REFEREE COMMENTS

Referee #1:

The present MS investigates the role of a micro-RNA (miR-138-5p) in excitatory synaptic transmission in the cerebellar cortex. This microRNA is strongly expressed in the cerebellar cortex and its role in regulating synaptic transmission in other brain regions has been revealed by previous studies, thus the authors addressed how sequestering miR-138-5p affects cerebellar mossy fibre (MF) to granule cell (GC) EPSCs. The experiments are of a high standard and the data is presented clearly. Most conclusions are supported by the data (but not the 'increase in AMPAR numbers at synapses').

The authors decided to investigate synaptic transmission between cerebellar MFs and GCs. In the first part of the MS, they demonstrate that miR-138-5p and Erbb4 are expressed in GCs. They, however, did not investigate their expression in the brainstem nuclei, from which MFs originate. This is important because they describe a presynaptic change in function. Is that because there is a change in the presynaptic cell or because there is an unknown retrograde signaling form the GCs. This issue can be also addressed by selective, virus-mediated expression of Cre-recombinase in GCs only.

Th reviewer assumes that the authors performed single fiber stimulations throughout the MS. It would be nice to show the step-like responses in the stimulus strength vs. EPSC amplitude plot.

Line 185: 'This suggests that miR-138-5p negatively regulates the number of synaptic AMPARs in cerebellar GCs.' This analysis reveals that the number of open AMPA receptors has changed. This could be the consequence of an increase in the number or the open probability of AMPARs.

Line 248: 'whereas release probability was increased in miR-138 sponge animals (Cohen's d: 0.44 [95%CI -0.07, 0.94], Figure 5H).' In the figure, a p value of 0.08 is shown, suggesting that it is reasonably likely that such a difference is the consequence of a random process.

Referee #2:

Delvendahl et al. performed electrophysiological investigation on the excitatory synaptic transmission between mossy fibre and cerebellar granule cells in slice preparation using an miR-185-5p-sequestering sponge expressing mutant mouse line. They found that the amplitude and decay of evoked EPSC and mEPSC were larger and slower, respectively, in the mutant. They further performed mEPSC variance analysis to deduce the single-channel conductance and the channel number of the AMPA-type glutamate receptor, then found that channel density at the post-synaptic side is lower in the mutant. On the

presynaptic side, increased release probability at the same synapse in the mutant was demonstrated by increased quantal content, decreased coefficient of variance of evoked EPSC, reduced paired pulse ratio, and high frequency train stimulation. They applied a short-term plasticity model to support their idea that the release probability was increased in the mutant. All of the experiments and analyses were precisely arranged and the analysis results are enough to convince readers their arguments. Therefore, the conclusion that miR-185-5p increases synaptic efficacy at the MF-GC synapse through both pre- and post-synaptic mechanisms is reasonable. However, the reviewer feel that the physiological impact of the findings is modest as opposed to the importance that authors emphasize: differences in amplitudes of the evoked and miniature EPSCs, about 37% and 17% of the average values, are not very large to persuade that the function of miR-185-5p is critically important at this synapse, though the differences are statistically significant. Such changes in the synaptic properties could naturally occur when such an miRNA that affects lifetime of wide range of mRNA species is lost throughout the development. It would have been more relevant if the sponge expression was controlled temporally and/or in cell-type or region specific manner. Because the range of target RNA of miR-185-5p is unclear, clear molecular schemes that take place under the loss of miR-185-5p cannot be drawn. And that the loss of miR-185-5p impacts the synaptic properties has already reported using the same sponge construct, but in inhibitory synapse, by the author's group (Daswani et al., 2022).

The number of animals seems to be not enough to conclude the differences between the genotypes. One of the mutant cells showed average evoked EPSC amplitudes around 260-270 pA (Fig. S2C). Considering the distribution of evoked EPSC amplitudes of all 4 mutant animals (Fig. 2D), the EPSC amplitude events of more than 200 pA may belong to the single cell of average 260-270 pA. If it is the case, the difference in evoked EPSC amplitude between the mutant and control may have been exaggerated by the single mutant animal that showed much larger EPSC amplitudes at the MF-GC synapse.

The number of mEPSC events of more than 20 pA shown in Fig. 3B (sponge mouse) seems to be 13 or more to say the least. Then, the largest population in the mEPSC amplitudes in the mutant (Fig. 3C) should belong to the single trace/animal shown in Fig. 3B, meaning that the difference in mEPSC amplitude between the genotypes was expanded by just one mutant animal that had much larger mEPSC amplitudes than other animals at the MF-GC synapse. The reviewer worries that the numbers of animals (especially the mutant) are too small to conclude the difference, even though the effect sizes are indicated. And the choice of the trace in Fig. 3B, as expressed as "Example recording", is not appropriate if the mEPSCs in the trace are the largest population in the mutant mEPSC group. Another trace with moderate mEPSCs in the mutant mEPSC distribution should be selected.

Minor points:

- In Fig. 3A and B, expanded traces to show some of the mEPSC wave forms would be useful.
- Mean values shown in bar graphs (e.g., Figs. 2D-G, 3C-F, 3I-J..) should be indicated as actual figures in figure legends or in corresponding parts of the main text, although the difference in the evoked EPSC can be found in the text as "~37%" (line 155 in the main text) or "36.9%" in line 200 and the difference of mEPSC as "17.0%" in line 200.
- Degrees of error should be indicated as figures together with figures of the mean values. Although error bars are shown in the graphs, it is unclear whether they indicate the standard deviation or the standard error of mean.
- Even if the properties of the granule cell-Purkinje cell synapse are mentioned in a supplemental figure (Fig. S3), more detail should be described. At least number and age of animals, conditions for recording mEPSC of granule cell-Purkinje cell synapse (holding potential, use of blockers for GABA/Glu receptors). It is not clear that the dots in Fig. S3A indicate each mEPSC or average of mEPSC in each animal. In the latter case, then the reviewer wonder why the authors do not show and compare every mEPSC event as they do in Fig. 2D. In Fig. S3B, number of experiments should be shown. And the EPSC I/O slopes of the granule cell/Purkinje cell synapse look different between the genotypes (bar graph shows >30% difference in the average values), but the authors do not mention about it and they concluded as "synaptic strength at parallel fibre to Purkinje cell synapses was not enhanced in miR-138 sponge mice" (line 161). Indeed, the statistical measure does not say the difference as "significant", it should be attributed to the small sample numbers.

END OF COMMENTS

EDITOR COMMENTS

Reviewing Editor:

This manuscript examines the role of micro-RNA, miR-138-5p, in regulating excitatory synaptic transmission at the mossy-fiber to granule synapses in the cerebellum. Both reviewers found the data that as highly rigorous and support the conclusions drawn. In addition, both felt that the data added to our understanding of synaptic transmission in the cerebellum. However, the reviewers had some concerns. To address the concern of whether the presynaptic phenotypes are observed are specific to the presynaptic cell or a postsynaptic change via retrograde signal, the authors should provide data on miR138-5p expression in the brainstem nuclei, or do selective expression of the miR138-5p in granule cells. In addition, there is concern about phenotypes observed may be exaggerated by a single mutant animal in the data set. The authors need to address this concern. The authors need to carefully revise and rewrite their manuscript give the positive and careful comments by the reviewers. In addition, the authors need to move all supplemental data into the main text to comply with Journal policy.

Please also see 'Required Items' above.

Authors' response: We would like to thank the editor for her/his assessment of the manuscript. We have extensively revised our manuscript, considering all comments of the two reviewers. In addition, we have included the most important parts of the supplementary figures in the main figures. Please find our point-by-point responses to the reviewers' comments below.

REFEREE COMMENTS

Referee #1:

The present MS investigates the role of a micro-RNA (miR-138-5p) in excitatory synaptic transmission in the cerebellar cortex. This microRNA is strongly expressed in the cerebellar cortex and its role in regulating synaptic transmission in other brain regions has been revealed by previous studies, thus the authors addressed how sequestering miR-138-5p affects cerebellar mossy fibre (MF) to granule cell (GC) EPSCs. The experiments are of a high standard and the data is presented clearly. Most conclusions are supported by the data (but not the 'increase in AMPAR numbers at synapses').

The authors decided to investigate synaptic transmission between cerebellar MFs and GCs. In the first part of the MS, they demonstrate that miR-138-5p and Erbb4 are

expressed in GCs. They, however, did not investigate their expression in the brainstem nuclei, from which MFs originate. This is important because they describe a presynaptic change in function. Is that because there is a change in the presynaptic cell or because there is an unknown retrograde signaling form the GCs. This issue can be also addressed by selective, virus-mediated expression of Cre-recombinase in GCs only.

Authors' response: We would like to thank the reviewer for bringing this point to our attention. Indeed, it is a highly relevant question if miR-138 is expressed in presynaptic neurons giving rise to cerebellar mossy fibres. We addressed this question using recently published sequencing data from the miRNATissueAtlas,¹ which revealed high expression of miR-138 in the two brain regions that are a major source of cerebellar mossy fibres, the pons and medulla oblongata (comparable to the cerebellum, see revised Figure 1). We also investigated the expression of *ErbB4* in precerebellar nuclei within the pons and medulla using single-nuclei RNA-seq data,² which revealed robust *ErbB4* expression in excitatory neurons of these nuclei (revised Figure 1). Although these results show that miR-138 and *ErbB4* are expressed in the brain areas from which cerebellar mossy fibres originate, this is obviously no direct evidence for pre-synaptic expression. Thus, it remains open if there is a direct effect on presynaptic function, or trans-synaptic regulation by miR-138, as discussed in the Discussion section (line 443 and 450). We hope to be able to address this interesting question in a future study.

The reviewer assumes that the authors performed single fiber stimulations throughout the MS. It would be nice to show the step-like responses in the stimulus strength vs. EPSC amplitude plot.

Authors' response: The reviewer correctly assumes that we recorded single inputs throughout the manuscript. We followed the procedures established in several previous publications^{3,4,5,6} to ensure single fibre stimulation during the search phase of EPSC responses. The stimulation strength for EPSCs was gradually decreased until

¹ S. Rishik et al., "miRNATissueAtlas 2025: An Update to the Uniformly Processed and Annotated Human and Mouse Non-Coding RNA Tissue Atlas," *Nucleic Acids Research* 53, no. D1 (2025): D129–37, <https://doi.org/10.1093/nar/gkae1036>.

² J. Langlieb et al., "The Molecular Cytoarchitecture of the Adult Mouse Brain," *Nature* 624, no. 7991 (2023): 333–42, <https://doi.org/10.1038/s41586-023-06818-7>.

³ R. A. Silver, S. G. Cull-Candy, and T. Takahashi, "Non-NMDA Glutamate Receptor Occupancy and Open Probability at a Rat Cerebellar Synapse with Single and Multiple Release Sites," *The Journal of Physiology* 494, no. Pt 1 (1996): 231–50, <https://doi.org/10.1113/jphysiol.1996.sp021487>.

⁴ S. Hallermann et al., "Bassoon Speeds Vesicle Reloading at a Central Excitatory Synapse," *Neuron* 68, no. 4 (2010): 710–23, <https://doi.org/10.1016/j.neuron.2010.10.026>.

⁵ I. Delvendahl, K. Kita, and M. Müller, "Rapid and Sustained Homeostatic Control of Presynaptic Exocytosis at a Central Synapse," *Proceedings of the National Academy of Sciences of the United States of America* 116, no. 47 (2019): 23783–89, <https://doi.org/10.1073/pnas.1909675116>.

⁶ K. Kita et al., "GluA4 Facilitates Cerebellar Expansion Coding and Enables Associative Memory Formation," *eLife* 10 (2021): e65152, <https://doi.org/10.7554/eLife.65152>.

a fixed intensity generated both successes and failures (Figure 1 below), as predicted for a single afferent.

Figure 1: Representative EPSCs recorded from a cerebellar granule cell after varying the stimulation intensity of the presynaptic mossy fibre input. Note the prominent difference in amplitudes, suggesting failures (black/grey), or the stimulation of a single afferent input (success, orange) after varying the stimulation intensity. Individual sweeps and averages are shown in light or dark, respectively.

The procedures for EPSC recordings are described in the Methods section (lines 551–554). We did not note down the (varying) stimulation intensity during the EPSC search. Hence, we unfortunately cannot display the suggested EPSC amplitude vs. stimulation intensity plot. However, we took care that the stimulation position and intensity were adjusted to stimulate a single afferent input only.

Line 185: 'This suggests that miR-138-5p negatively regulates the number of synaptic AMPARs in cerebellar GCs.' This analysis reveals that the number of open AMPA receptors has changed. This could be the consequence of an increase in the number or the open probability of AMPARs.

Authors' response: We would like to thank the reviewer for raising this point about the interpretation of mEPSC variance analysis results. Indeed, peak-scaled non-stationary fluctuation analysis reports the number of channels open at the peak of the synaptic current. We have changed our interpretation of the variance-mean analysis accordingly and have altered the respective statements in the key points, abstract, and main body of the manuscript. In addition, we have added a discussion of open probability and number changes underlying the observed effect (lines 401–405).

Line 248: 'whereas release probability was increased in miR-138 sponge animals (Cohen's d : 0.44 [95%CI -0.07, 0.94], Figure 5H).' In the figure, a p value of 0.08 is shown, suggesting that it is reasonably likely that such a difference is the consequence of a random process.

Authors' response: Throughout our study, we chose to report effect sizes (Cohen's d) and results of permutation t -tests.⁷ For release probability, there was a medium effect size (Cohen's d = 0.44), and the p -value indicates a low probability of obtaining

⁷ J. Ho et al., "Moving beyond P Values: Data Analysis with Estimation Graphics," *Nature Methods* 16, no. 7 (2019): 565–66, <https://doi.org/10.1038/s41592-019-0470-3>.

this result by chance. We would prefer to not make dichotomous interpretations of results based on a rather arbitrary threshold for p-values. Nevertheless, we now acknowledge a “slight” increase in release probability in the description of revised Figure 6 (line 314). Of note, several lines of evidence suggest the modulation of release probability in our experiments (Figures 5, 7).

Referee #2:

Delvendahl et al. performed electrophysiological investigation on the excitatory synaptic transmission between mossy fibre and cerebellar granule cells in slice preparation using an miR-185-5p-sequestering sponge expressing mutant mouse line. They found that the amplitude and decay of evoked EPSC and mEPSC were larger and slower, respectively, in the mutant. They further performed mEPSC variance analysis to deduce the single-channel conductance and the channel number of the AMPA-type glutamate receptor, then found that channel density at the post-synaptic side is lower in the mutant. On the presynaptic side, increased release probability at the same synapse in the mutant was demonstrated by increased quantal content, decreased coefficient of variance of evoked EPSC, reduced paired pulse ratio, and high frequency train stimulation. They applied a short-term plasticity model to support their idea that the release probability was increased in the mutant. All of the experiments and analyses were precisely arranged and the analysis results are enough to convince readers their arguments. Therefore, the conclusion that miR-185-5p increases synaptic efficacy at the MF-GC synapse through both pre- and post-synaptic mechanisms is reasonable. However, the reviewer feel that the physiological impact of the findings is modest as opposed to the importance that authors emphasize: differences in amplitudes of the evoked and miniature EPSCs, about 37% and 17% of the average values, are not very large to persuade that the function of miR-185-5p is critically important at this synapse, though the differences are statistically significant. Such changes in the synaptic properties could naturally occur when such an miRNA that affects lifetime of wide range of mRNA species is lost throughout the development. It would have been more relevant if the sponge expression was controlled temporally and/or in cell-type or region specific manner. Because the range of target RNA of miR-185-5p is unclear, clear molecular schemes that take place under the loss of miR-185-5p cannot be drawn. And that the loss of miR-185-5p impacts the synaptic properties has already reported using the same sponge construct, but in inhibitory synapse, by the author's group (Daswani et al., 2022).

Authors' response:

Thank you for the constructive feedback on our study. Regarding the interpretation and relevance of our findings, we would like to address the concerns point-by-point:

Relevance of findings

We believe that the observed alterations in synaptic efficacy upon miR-138 downregulation are of physiological relevance for the following reasons: (i) The increase in EPSC amplitude (37%) and charge (44%) are substantial and robust. AN increase in EPSC charge of this magnitude is expected to have a profound impact on GC spiking,⁸ and thus has the potential to broadly modulate information processing in the cerebellar cortex. Indeed, simulations indicate that the observed changes in EPSC properties upon miR-138 downregulation would increase GC Spiking upon high-frequency MF input by about 50%. This would likely impair GC function, which are responsible for generating a sparse population code.^{9,10,11} (ii) The magnitude of synaptic strength changes in our study are in the range of LTP reported at the MF–GC synapse,^{12,13} again suggesting that this degree of synaptic modification has a physiological impact. (iii) We are not aware of any previous study reporting enhanced MF–GC synaptic strength upon a genetic manipulation.

Nevertheless, we take the reviewer’s comment into account and have changed several statements in the abstract, introduction, and discussion to better reflect the potential impact of our findings.

Developmental compensations

We agree that the continuous expression of the sponge construct might cause multiple changes during development, which could contribute to the observed effects. This is now discussed in the revised manuscript (lines 436–440).

miR-138 targets

The reviewer makes an astute point that knowing the mRNA target(s) in the cerebellum would greatly enhance our understanding of miR-138’s role. Although *Erb4* is an intriguing candidate, we cannot provide any direct evidence for this gene being involved in the observed synaptic changes. Given the wide range of potential (synaptic) targets for microRNAs, we believe that identifying the exact target(s) is beyond the scope of the present study.

Comparison to previous work on miR-138

Indeed, Daswani et al. reported a change in inhibitory transmission in the hippocampus upon downregulation of miR-138, but excitatory transmission was not altered.¹⁴

⁸ Kita et al., “GluA4 Facilitates Cerebellar Expansion Coding and Enables Associative Memory Formation.”

⁹ D. Marr, “A Theory of Cerebellar Cortex,” *The Journal of Physiology* 202, no. 2 (1969): 437–70, <https://doi.org/10.1113/jphysiol.1969.sp008820>.

¹⁰ J. S. Albus, “A Theory of Cerebellar Function,” *Mathematical Biosciences* 10, no. 1–2 (1971): 25–61, [https://doi.org/10.1016/0025-5564\(71\)90051-4](https://doi.org/10.1016/0025-5564(71)90051-4).

¹¹ G. Billings et al., “Network Structure within the Cerebellar Input Layer Enables Lossless Sparse Encoding,” *Neuron* 83 (2014): 960–74, <https://doi.org/10.1016/j.neuron.2014.07.020>.

¹² E. D’Angelo et al., “Evidence for NMDA and mGlu Receptor-Dependent Long-Term Potentiation of Mossy Fiber-Granule Cell Transmission in Rat Cerebellum.,” *Journal of Neurophysiology* 81, no. 1 (1999): 277–87.

¹³ M. Sgritta et al., “Hebbian Spike-Timing Dependent Plasticity at the Cerebellar Input Stage,” *Journal of Neuroscience* 37, no. 11 (2017): 2809–23, <https://doi.org/10.1523/JNEUROSCI.2079-16.2016>.

¹⁴ R. Daswani et al., “MicroRNA-138 Controls Hippocampal Interneuron Function and Short-Term Memory in Mice,” *eLife* 11 (2022): e74056, <https://doi.org/10.7554/eLife.74056>.

Our findings in the cerebellum differ from this previous study in several ways. (i) We could demonstrate that miR-138 downregulation affects excitatory transmission. (ii) We observed marked changes in both miniature and evoked transmission. (iii) The enhanced synaptic strength is due to both pre- and postsynaptic alterations. (iv) We uncovered a role for miR-138 outside of the hippocampus, in the cerebellum. (v) We provide evidence for a regulation of excitatory synapses by miR-138 in the cerebellum, whereas this miRNA controls inhibitory synaptic transmission in the hippocampus, thus suggesting a brain-region-specific role of miR-138.

The number of animals seems to be not enough to conclude the differences between the genotypes. One of the mutant cells showed average evoked EPSC amplitudes around 260-270 pA (Fig. S2C). Considering the distribution of evoked EPSC amplitudes of all 4 mutant animals (Fig. 2D), the EPSC amplitude events of more than 200 pA may belong to the single cell of average 260-270 pA. If it is the case, the difference in evoked EPSC amplitude between the mutant and control may have been exaggerated by the single mutant animal that showed much larger EPSC amplitudes at the MF-GC synapse.

Authors' response: We apologize for the lack of clarity in the previous version of the manuscript. All plots in the main figures display single data points from individual cells. For example, the EPSC amplitude comparison includes n=48 granule cells for control (from N=5 mice), and n=38 for the sponge condition (from N=4 mice). Each data point is the average of 12–36 EPSCs per cell, and each animal contributed at least n=8 cells to the analysis. In the revised manuscript, the number of cells and animals is now clearly stated in the figure legends.

In addition, we noted a mistake in the per-animal analyses presented in the supplementary figures, which were supposed to show the averages per animal as data points but were inadvertently showing a random cell per animal instead. We corrected this error and re-analysed the data on a per-animal basis. The corrected analysis revealed large effect sizes for EPSC amplitude (Cohen's $d = 2.73$ [95%CI 1.32, 4.41]), mEPSC amplitude (Cohen's $d = 1.95$ [95%CI 0.96, 2.89]), and PPR (Cohen's $d = -1.49$ [95%CI -3.10, 0.29]). However, because the Journal of Physiology does not feature supplementary figures, and because we do not consider the per-animal analysis as essential for the paper, these results are no longer shown in the revised manuscript.

The number of mEPSC events of more than 20 pA shown in Fig. 3B (sponge mouse) seems to be 13 or more to say the least. Then, the largest population in the mEPSC amplitudes in the mutant (Fig. 3C) should belong to the single trace/animal shown in Fig. 3B, meaning that the difference in mEPSC amplitude between the genotypes was expanded by just one mutant animal that had much larger mEPSC amplitudes than other animals at the MF-GC synapse. The reviewer worries that the numbers of

animals (especially the mutant) are too small to conclude the difference, even though the effect sizes are indicated. And the choice of the trace in Fig. 3B, as expressed as "Example recording", is not appropriate if the mEPSCs in the trace are the largest population in the mutant mEPSC group. Another trace with moderate mEPSCs in the mutant mEPSC distribution should be selected.

Authors' response: As stated in response to the previous point, the quantifications in the figures are showing averages of individual cells and we apologize for this being not clear in the previous manuscript.

For mEPSC quantification, we recorded from n=53 cells in N=5 control mice, and n=41 cells in N=4 sponge mice. This represents a relatively large sample size for electrophysiological studies. Because each animal, regardless of genotype, contributed at least n=8 cells, our results are unlikely dominated by a single animal or a single cell.

According to the suggestion of the reviewer, we have changed the sponge example in Figure 3 (now Figure 4 in the revised manuscript). The mEPSC examples shown are now more representative of the two groups. In addition, we provide the number of cells and animals in all figures.

Minor points:

- In Fig. 3A and B, expanded traces to show some of the mEPSC wave forms would be useful.

Authors' response: Thank you for this suggestion. We have included the average mEPSC waveform of the two example recordings shown in Figure 4 of the revised manuscript.

- Mean values shown in bar graphs (e.g., Figs. 2D-G, 3C-F, 3I-J..) should be indicated as actual figures in figure legends or in corresponding parts of the main text, although the difference in the evoked EPSC can be found in the text as "~37%" (line 155 in the main text) or "36.9%" in line 200 and the difference of mEPSC as "17.0%" in line 200.

Authors' response: We have added the mean values for comparisons of control and sponge throughout the Results section.

- Degrees of error should be indicated as figures together with figures of the mean values. Although error bars are shown in the graphs, it is unclear whether they indicate the standard deviation or the standard error of mean.

Authors' response: Thank you for highlighting this oversight. The error bars represent the 95% confidence interval, and we have included this information in the Methods section and all figure legends.

- Even if the properties of the granule cell-Purkinje cell synapse are mentioned in a supplemental figure (Fig. S3), more detail should be described. At least number and age of animals, conditions for recording mEPSC of granule cell-Purkinje cell synapse (holding potential, use of blockers for GABA/Glu receptors). It is not clear that the dots in Fig. S3A indicate each mEPSC or average of mEPSC in each animal. In the latter case, then the reviewer wonder why the authors do not show and compare every mEPSC event as they do in Fig. 2D. In Fig. S3B, number of experiments should be shown. And the EPSC I/O slopes of the granule cell/Purkinje cell synapse look different between the genotypes (bar graph shows >30% difference in the average values), but the authors do not mention about it and they concluded as "synaptic strength at parallel fibre to Purkinje cell synapses was not enhanced in miR-138 sponge mice" (line 161). Indeed, the statistical measure does not say the difference as "significant", it should be attributed to the small sample numbers.

Authors' response: We would like to thank the reviewer for pointing out the omission of methods description for the PC recordings. We have included this information in the revised manuscript. PC recordings were performed in the same animals as GC recordings (n=5 and n=4 mice for control and sponge, respectively). As in all the figures, the data points represent the average of mEPSCs or EPSCs for a given PC recording. The figure legend now includes the number of cells and animals.

Regarding the interpretation of the PF-PC EPSC slope, we agree with the reviewer that the sample size is rather small in this case. There is significant overlap between the two distributions, and we therefore only concluded that PF-PC transmission does not appear to be enhanced. With the currently limited data, it is difficult to assess whether there is a reduction in the I/O slope. This is now highlighted in the text (line 201–202).

END OF COMMENTS

Dear Dr Delvendahl,

Re: JP-RP-2025-288019R1 "microRNA-138-5p suppresses excitatory synaptic strength at the cerebellar input layer" by Igor Delvendahl, Reetu Daswani, Jochen Winterer, Pierre-Luc Germain, Nora Maria Uhr, Gerhard Schratt, and Martin Mueller

We are pleased to tell you that your paper has been accepted for publication in The Journal of Physiology.

Yours sincerely,

Katalin Toth
Senior Editor
The Journal of Physiology

If you would like to receive our 'Research Roundup', a monthly newsletter highlighting the cutting-edge research published in The Physiological Society's family of journals (The Journal of Physiology, Experimental Physiology, Physiological Reports, The Journal of Nutritional Physiology and The Journal of Precision Medicine: Health and Disease), please click this link, fill in your name and email address and select 'Research Roundup':

<https://www.physoc.org/journals-and-media/membernews>

- **TRANSPARENT PEER REVIEW POLICY:** To improve the transparency of its peer review process, The Journal of Physiology publishes online as supporting information the peer review history of all articles accepted for publication. Readers will have access to decision letters, including Editors' comments and referee reports, for each version of the manuscript as well as any author responses to peer review comments. Referees can decide whether or not they wish to be named on the peer review history document.
- You can help your research get the attention it deserves! Check out Wiley's free Promotion Guide for best-practice recommendations for promoting your work at: www.wileyauthors.com/eeo/guide. You can learn more about Wiley Editing Services which offers professional video, design, and writing services to create shareable video abstracts, infographics, conference posters, lay summaries, and research news stories for your research at: www.wileyauthors.com/eeo/promotion.
- **IMPORTANT NOTICE ABOUT OPEN ACCESS:** To assist authors whose funding agencies mandate public access to published research findings sooner than 12 months after publication, The Journal of Physiology allows authors to pay an Open Access (OA) fee to have their papers made freely available immediately on publication.

EDITOR COMMENTS

Reviewing Editor:

The authors have done an excellent job of responding to the reviewers' previous critiques. There are no further concerns.

REFEREE COMMENTS

Referee #1:

All arguments accepted.

Referee #2:

In this revision, the authors have fully addressed my concerns.